# Insights into the operational stability of wide-bandgap perovskite and tandem solar cells under rapid thermal cycling

Kun Sun [1,12], Renjun Guo [2,3] ✉, Qilin Zhou [4,5], Lingyi Fang [2,3], Xiongzhuo Jiang [1], Simon A. Wegener [1], Yuxin Liang[1], Zerui Li[1], Suzhe Liang[6,7], Matthias Schwartzkopf[8], Erkan Aydin[9], Sarathlal Koyiloth Vayalil [8,10], Stephan V. Roth [8,11], Ulrich W. Paetzold [2,3] & Peter Müller-Buschbaum [1] ✉

Temperature variations can induce phase transformations and strain in perovskite solar cells (PSCs), undermining their structural stability and device performance. Despite growing interest, the operational stability of triple-cation wide-bandgap (WBG) PSCs and tandem solar cells (TSCs) under rapid solar-thermal cycling remains poorly understood. Here, we investigate the operational stability of WBG PSCs (~1.68 eV) with a champion power conversion efficiency (PCE) of 24.31% and extend the study to TSCs. We find that degradation during device operation under rapid solar-thermal cycling (temperature change rate of 10 °C/min) is independent of passivation and occurs in two distinct regimes: an initial burn-in phase, which accounts for a rapid 60% relative loss in performance, followed by a steady degradation characterized by temperature-dependent fluctuations in photovoltaic parameters. By *operando* grazing-incidence wide-angle X-ray scattering and photoluminescence measurements, we reveal that temperature-induced strain, phase transition, and the increased non-radiative recombination collectively contribute to the degradation of PSCs. This work advances the understanding of the degradation mechanisms of WBG PSCs and TSCs, providing insights toward improving their operational thermal stability for real-world applications.

Single-junction perovskite solar cells (PSCs) have achieved remarkable progress in recent years, with the certified power conversion efficiency (PCE) reaching 27.3%[1]. To further overcome the Shockley-Queisser limit, tandem perovskite solar cells (TSCs), by integrating wide-bandgap perovskite solar cells (1.63 eV–1.80 eV) with other solar cells, e.g., crystalline silicon, CIGS, and narrow-bandgap PSCs are utilized[2–9], and their certified PCE is now reaching 34.85%[10]. In terms of the wide-bandgap PSCs for tandem application, a higher bromide content is

[1]Technical University of Munich, TUM School of Natural Sciences, Department of Physics, Chair for Functional Materials, James-Franck-Str. 1, Garching, Germany. [2]Karlsruhe Institute of Technology (KIT), Institute of Microstructure Technology, Herrmann-von-Helmholtz-Platz 1, Karlsruhe, Germany. [3]Karlsruhe Institute of Technology (KIT), Light Technology Institute, Engesserstr. 13, Karlsruhe, Germany. [4]National University of Singapore, Department of Chemical and Biomolecular Engineering, Singapore, Singapore. [5]National University of Singapore, Solar Energy Research Institute of Singapore (SERIS), Singapore, Singapore. [6]Eastern Institute for Advanced Study, Ningbo Institute of Digital Twin, Eastern Institute of Technology, Ningbo, China. [7]Zhejiang Key Laboratory of All-Solid-State Battery, Ningbo Key Laboratory of All-Solid-State Battery, Ningbo, China. [8]Deutsches Elektronen-Synchrotron, Notkestr. 85, Hamburg, Germany. [9]Ludwig-Maximilians-Universität München (LMU), Department of Chemistry, Butenandtstr. 11, München, Germany. [10]Applied Sciences Cluster, University of Petroleum and Energy Studies UPES, Dehradun, Uttarakhand, India. [11]Royal Institute of Technology KTH, Department of Fibre and Polymer Technology, Teknikringen 56-58, Stockholm, Sweden. [12]Present address: Helmholtz-Zentrum Berlin für Materialien und Energie GmbH, Hahn-Meitner-Platz 1, Berlin, Germany. ✉e-mail: renjun.guo@kit.edu; muellerb@ph.tum.de

required to realize the desired bandgap[11,12]. Nevertheless, a higher bromide content in perovskites normally leads to a rapid crystallization process, which in turn results in higher defect densities accompanied by structure and composition inhomogeneity and a high density of grain boundaries[13–15]. In addition, compared with their mid-bandgap counterparts (1.5 eV–1.63 eV), wide-bandgap PSCs suffer from phase separation under light illumination and non-radiative recombination at perovskite/charge transporting layers (CTLs)[6,16,17], leading to a large open-circuit voltage ($V_{OC}$) deficit (defined as $\frac{E_g}{q} - V_{OC}$, where $E_g$ and $q$ refer to bandgap and elementary charge, respectively). Universal methods, such as forming a 2D capping layer with bulky organic cations, e.g., phenylethylammonium (PEA$^+$) and butylammonium (BA$^+$)[18–20], using Lewis bases (e.g., short-chain diammonium ligands)[10,18,21–24], adding an ultrathin LiF or MgF$_x$ interlayer[10,25,26], and additive engineering[2,27], have been proven effective in reducing the $V_{OC}$ losses and inhibit phase segregation in wide-bandgap PSCs.

In addition, to understand the behavior of the perovskite-based solar cells for terrestrial and extraterrestrial applications[28–30], their thermal cycling performance needs to be evaluated. Previous work demonstrated that perovskite and its respective CTLs endure alternating tension and compression under thermal cycling, which could result in ion migration, phase transition, crystal disorder, delamination, and device failure[31–34]. However, these pioneering studies have primarily focused on the mid-bandgap PSCs (around 1.5 eV) under thermal cycling[32,34,35], leaving the thermal and operational stability of WBG PSCs and perovskite/Si TSCs, especially under thermal cycling, largely unexplored. In particular, the cycling duration is not specified in the International Summit on Organic Photovoltaic Stability (ISOS) protocols for thermal cycling (ISOS-T) and solar-thermal cycling (ISOS-LT). These protocols define only parameters such as temperature range, environmental atmosphere, and light source[29]. In this regard, cycling durations have varied significantly in thermal cycling stability tests reported in the literature[34,36]. In addition, the International Electrotechnical Commission (ICE) standard test procedure specifies a cycle duration of 3 to 6 h, with a maximum temperature change rate of 100 °C/h[37,38]. However, such a low rate is not suitable for *operando* studies or rapid prototyping of perovskite composition, passivation, and device structures. Therefore, we define a full thermal cycle with a temperature change rate of ~10 °C/min as rapid thermal cycling.

In this work, a champion PCE of 24.31% is achieved in 0.05-cm$^2$ WBG PSCs (1.68 eV) with dual passivation using 3-fluorophenethylamine iodide (3-F-PEAI) and ethylenediamine diiodide (EDAI$_2$). We first determine the temperature coefficients of single-junction WBG PSCs with and without dual passivation and perovskite/Si TSCs at different temperature ranges. We note that the thermal coefficients of single-junction PSCs and TSCs extracted from $J$–$V$ measurements at constant temperatures do not reflect device performance under thermal cycling conditions. We further investigate the solar cell behavior of WBG PSCs under rapid solar-thermal cycling conditions, while concurrently analyzing their structural evolution by GIWAXS. We find that the degradation of WBG PSCs universally follows two characteristic regimes, an initial burn-in phase followed by steady degradation, regardless of their initial performance or passivation strategy. The degradation behavior is driven by the accumulation of non-radiative recombination centers, temperature-induced phase transitions, and strain, collectively leading to pronounced losses in fill factor (*FF*) and $V_{OC}$. When integrated with Si bottom cells, TSCs demonstrate an improved temperature resilience at low temperatures and retain 94% of their original PCE after over 200 min of thermal cycling.

## Results

### Photovoltaic performance

First, we investigate the effects of single and dual passivation on the properties of perovskite films. Hereafter, the samples without passivation, with EDAI$_2$ passivation, and with 3-F-PEAI and EDAI$_2$ dual passivation are referred to as control, EDAI$_2$ and DP, respectively. The photoluminescence (PL) spectra (Supplementary Fig. 1a) show that the passivation does not alter the bandgap of the perovskite. As demonstrated in previous work, EDAI$_2$ shifts the Fermi level closer to the conduction band minimum, leading to stronger n-type doping and effective field-effect passivation after treatment[10,21,22,24], whereas F-PEAI is shown to form a 2D capping layer (Supplementary Fig. 2) and suppress the defective centers of the perovskite surface[6,39], contributing to a better device performance. Together, these passivation strategies enhance film quality without altering optical absorption, as confirmed by the unchanged absorbance spectra (Supplementary Fig. 1b), highlighting their promise for defect suppression and interface optimization.

To further characterize the chemical composition of the respective perovskite thin film, Fourier transform infrared (FTIR) spectra are performed (Supplementary Fig. 1c), where the molecular vibration features of C–N and C–F asymmetric stretching signals $v_{as}$ (C–N) and $v_{as}$ (C–F) locate at 1715 cm$^{-1}$ and 1120 cm$^{-1}$, respectively, indicating the successful incorporation of 3-F-PEAI into perovskite thin film. The root-mean-square (RMS) roughness of perovskite thin film after passivation slightly decreases in the order of EDAI$_2$ (13.27 nm) and DP (14.53 nm) thin films compared to that of control (16.25 nm) film, as evidenced by atomic force spectroscopy (AFM) measurements (Supplementary Fig. 3). In addition, in situ PL is utilized to assess the phase stability in response to light-induced stress (Supplementary Fig. 4). We find a slight red shift in the control thin film over time, while EDAI$_2$ and DP thin films demonstrate better stability under light illumination (with no visible change in peak positions), showing the passivation effect for suppressing photo-induced phase segregation and potentially improving device operational stability.

Next, we fabricate solar cells with the configuration of ITO/NiO$_x$/self-assembled monolayers (SAMs, Me-4PACz)/WBG PSCs (750 nm, Cs$_{0.05}$MA$_{0.1}$FA$_{0.85}$PbI$_{0.77}$Br$_{0.23}$)/ passivation layer/C$_{60}$/BCP/Ag (Fig. 1a, detailed fabrication process can be found in the Experimental section). The champion device performance for all three types of PSCs is shown in Fig. 1b, Supplementary Fig. 5, where the control device demonstrates a champion PCE of 20.38%, an FF of 76.83%, a short-circuit current ($J_{SC}$) of 22.29 mA/cm$^2$, and a $V_{OC}$ of 1.19 V. In comparison, the device with dual passivation displays superior performance, achieving a PCE of 24.31%, a $J_{SC}$ of 22.34 mA/cm$^2$, a $V_{OC}$ of 1.29 V, and an FF of 84.36%. In addition, the statistical photovoltaic parameters for all three PSCs types are compared (Supplementary Fig. 6), in which the DP devices demonstrate the highest $V_{OC}$ (average of 1.279 V) and FF (average of 79.68%), suggesting reduced non-radiative recombination and better energy alignment after passivation, ultimately contributing to the highest PCE among all the devices. Furthermore, the integrated $J_{SC}$ extracted from the external quantum efficiency (EQE, Fig. 1c, Supplementary Fig. 7) measurements is 21.2, 21.0, 21.2 mA/cm$^2$, showing less than a 5% mismatch compared to those obtained from the $J$–$V$ measurements. (Fig. 1b). The corresponding band gaps extracted from the EQE spectra are approximately 1.69 eV (Supplementary Fig. 8). Moreover, the quasi-steady-state measurement (steady state PCEs) of control, EDAI$_2$, and DP devices are 17.74, 21.50, and 21.96%, respectively (Fig. 1d, Supplementary Fig. 9).

Despite standard testing conditions (STC) specifying a temperature of 25 °C for measuring the device performance, the actual device temperature can reach 65 °C under operational conditions[40]. Furthermore, for further deployment of perovskite/silicon tandem solar modules in an outdoor environment, understanding device performance across a range of temperatures is essential for predicting solar cell and panel performance. The temperature coefficient ($\gamma$), which is equal to the change in PCE relative to PCE at room temperature divided

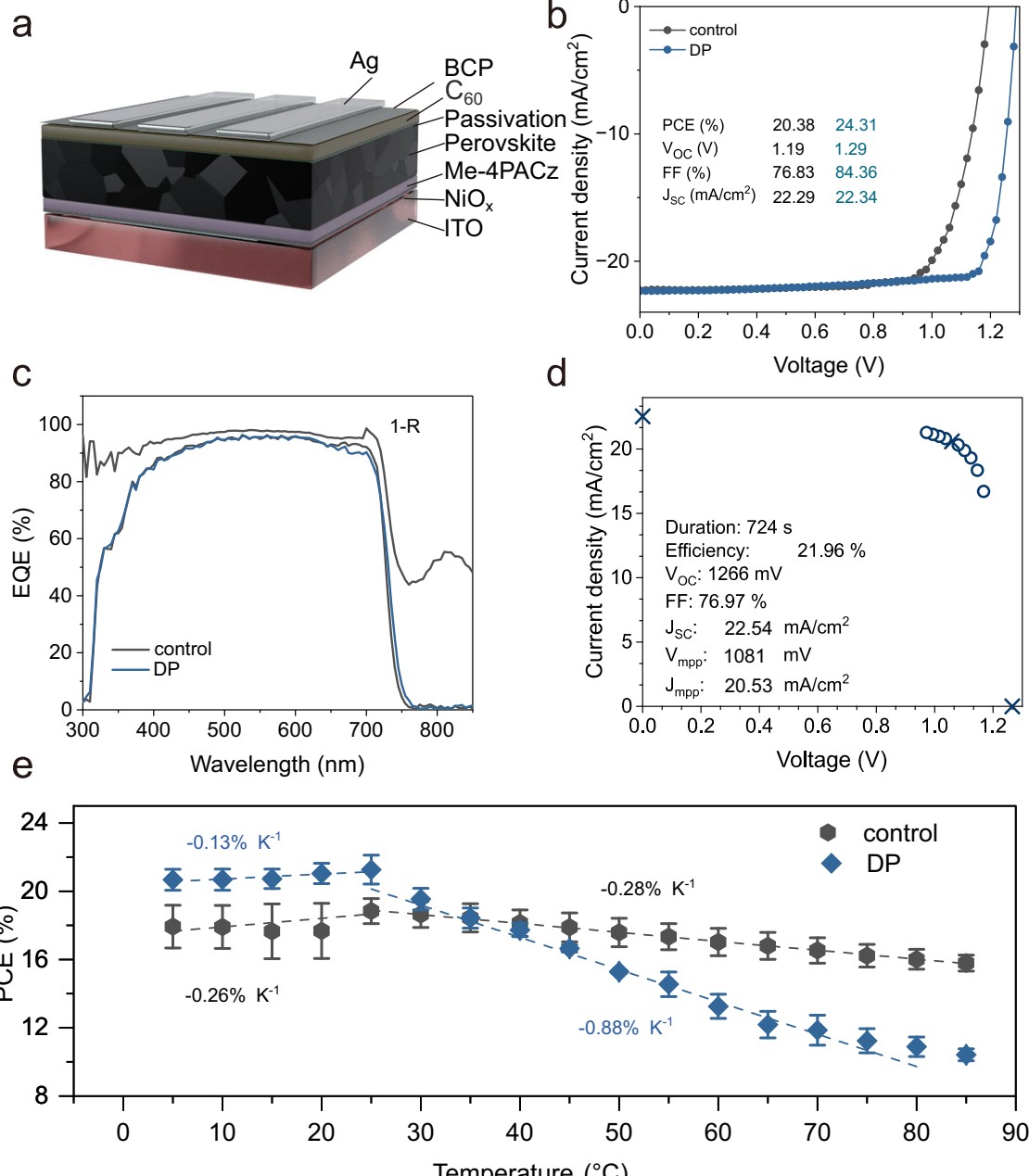

**Fig. 1 | Device performance and its temperature coefficients. a** Device config-uration with the main parts labeled. **b** *J–V* curves of the respective champion device under AM 1.5 G illumination equivalent to 100 mW cm⁻², retrieved from the reverse scan. **c** EQE spectra of the corresponding champion device and total absorbance (1-R) of the champion device with EDAI₂ and 3-F-PEAI dual passivation. **d** The quasi-steady-state measurement of the champion device with EDAI₂ and 3-F-PEAI dual passivation. **e** PCE as a function of temperature, where the error bar indicates the standard deviation extracted from 8 pixels.

by the change in temperature during device operation at varying temperature[40,41], can be calculated as:

$$\gamma = \frac{PCE_T - PCE_{RT}}{|(T - T_{RT})| \times PCE_{RT}} \qquad (1)$$

where $PCE_T$ and $PCE_{RT}$ correspond to PCE at varied temperature $T$ and room temperature, $T_{RT}$, respectively. As such, to calculate $\gamma$, the device performance is measured at different temperatures ranging from 5 to 85 °C in 5 °C intervals (Fig. 1e). The experimental setup for temperature-dependent *J–V* measurements is described in detail in the Experimental section and shown in Supplementary Fig. 10. We determine the temperature coefficients by extracting the slope from

the linear fits of PCE as a function of temperature and normalizing it to the PCE at room temperature (25 °C). Notably, the temperature coefficients in the 5–25 °C range ($\gamma_{LT}$) and the 25–80 °C ($\gamma_{HT}$) are largely different. At higher temperatures (above room temperature), however, DP devices suffer from a faster degradation, likely due to dimensionality collapse, penetration of bulky organic cation into the 3D perovskite layer within 2D/3D heterostructure, which hinders charge transfer[42–44]. The calculated $\gamma_{HT}$ values of control and DP devices are −0.28% K⁻¹ and −0.88% K⁻¹ respectively, comparable to −0.29% K⁻¹ of SunPower's silicon modules, -0.39% K⁻¹ of a standard monocrystalline module, and the thermal coefficients reported in the literature (Supplementary Table 1), while slightly lower than −0.17% K⁻¹ of a triple-cation PSC (Cs₀.₀₅(FA₀.₈₃MA₀.₁₇)Pb₁.₁(I₀.₈₃Br₀.₁₇)₃)[40,41]. This

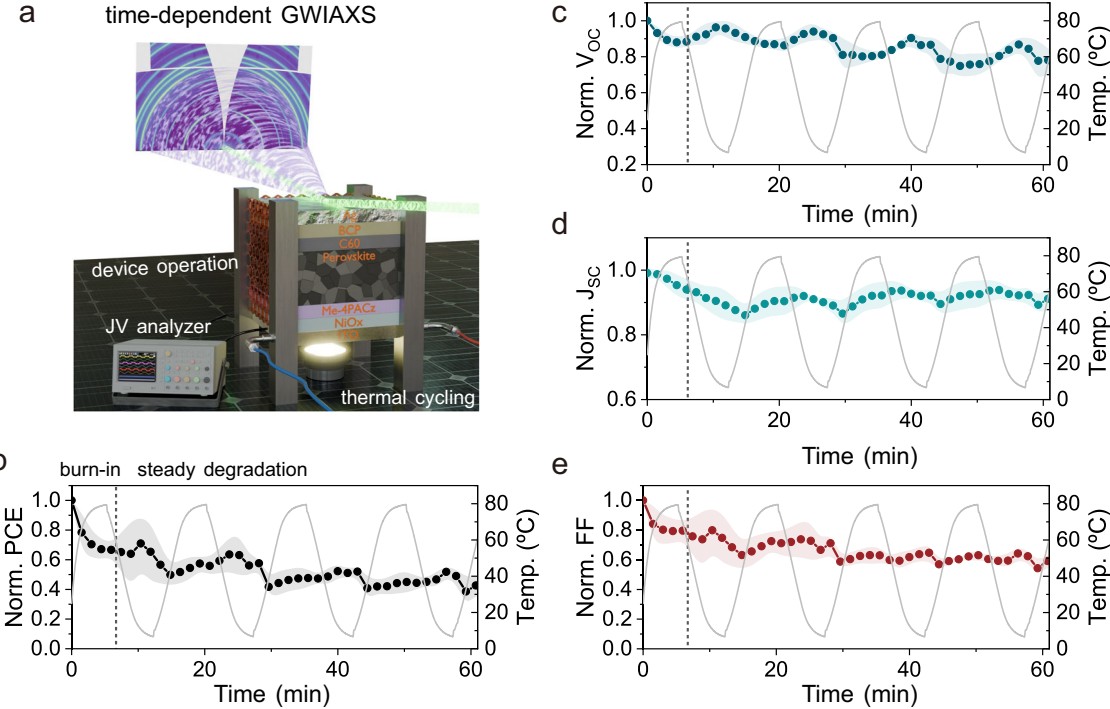

**Fig. 2 | Evolution of the device performance under solar-thermal cycling conditions. a** Schematic illustration of *operando* GIWAXS measurements during device operation under rapid solar-thermal cycling conditions. Normalized photovoltaic parameters as a function of time and temperature (indicated by the gray curve, showing the thermal cycling between 5 and 80 °C with each individual cycle of 15 min and a temperature change rate of -10 °C/min). The *J–V* measurements were performed at 1 min intervals, and the shaded areas refer to error bars derived from the standard deviation of respective photovoltaic parameters of five pixels subjected to rapid solar-thermal cycling. **b** Normalized PCE and temperature *versus* time, **c** normalized $V_{OC}$ and temperature *versus* time, **d** normalized $J_{SC}$ and temperature *versus* time, **e** normalized FF and temperature *versus* time.

difference in temperature coefficient might arise from many factors or a combination of them, e.g., perovskite composition, device configuration, measurement conditions, and light-soaking effects. However, identifying the exact cause of this difference is not the main scope of this work, since device behavior differs significantly under static temperature-dependent *J–V* measurement and rapid thermal cycling conditions. In addition, the $\gamma_{LT}$ values are −0.26% K$^{-1}$ and −0.13% K$^{-1}$ for the control and DP device, respectively, showcasing better stability at low temperatures with dual passivation. Overall, the metastability of high-efficiency WBG PSCs under non-standard temperature conditions suggests that current passivation strategies may not be suitable for operation under extreme conditions or may not necessarily contribute to enhanced stability, highlighting the need to develop dedicated strategies specifically aimed at improving stability.

## Operational stability under rapid thermal cycling

To concurrently track the structural evolution and changes in device performance of WBG PSCs under rapid thermal cycling conditions, *operando* GIWAXS measurements are performed during device operation, as mapped out in Fig. 2a. As the degradation behavior is quite similar in all types of PSCs, only the evolution of photovoltaic parameters for devices with dual passivation is shown in Fig. 2b–e, Supplementary Fig. 11. Overall, the average PCE of DP devices decreases to 42% of its original PCE after 1 h of solar-thermal cycling, originating from the combined degradation of $J_{SC}$ (9%), $V_{OC}$ (22%), and FF (41%). Similarly, the control devices (Supplementary Fig. 12) follow a comparable trend, with their average PCE decreasing to 51% of the original value, accompanied by reductions in $J_{SC}$ (12%), $V_{OC}$ (15%), and FF (32%) values. The FF fluctuations (characterized by the series resistance in the limit of infinitely large shunting resistance, Supplementary Fig. 13) can be correlated with the high defect densities and

the asymmetric strain experienced during rapid solar-thermal cycling, which will be discussed in the following sections. The timescale of light illumination and electrical bias, together with the temperature change rate, likely contribute collectively to the different degradation rates observed in devices under rapid solar thermal cycling compared with those under static temperature conditions. By simulating the PCE using the extracted coefficients and comparing it with the measured PCE over time and temperature (Supplementary Fig. 14), we decouple degradation mechanisms from temperature-induced performance changes under rapid solar thermal cycling. We classify the degradation process into two regimes by their degradation rates: the initial burn-in regime (approximately up to the first PCE recovery, Supplementary Fig. 14), which contributes to more than 60% of the total degradation in all three types of PSCs, and the steady degradation regime, where photovoltaic parameters partially recover, stabilize, and align well with temperature evolution. In addition, $V_{OC}$ and FF are the most affected parameters during solar-thermal cycling. Breaking down the degradation of respective photovoltaic parameters, we find that $V_{OC}$ follows an opposite trend with respect to temperature evolution, reaching its maximum at the lowest temperature. In principle, a higher temperature leads to an enhanced intrinsic charge carrier concentration, resulting in an increase in the dark saturation current ($J_0$) and, therefore, a decrease in $V_{OC}$ (Eq. (2)).

$$V_{OC} = \frac{nk_BT}{q}ln\left(\frac{J_{SC}}{J_0}+1\right) \quad (2)$$

Where $q$, $n$, and $k_B$ refer to the elementary charge, the ideality factor, and Boltzmann's constant, respectively. $J_{SC}$ exhibits a similar trend following the temperature profile, which can be attributed to temperature-induced bandgap broadening or narrowing. However, at

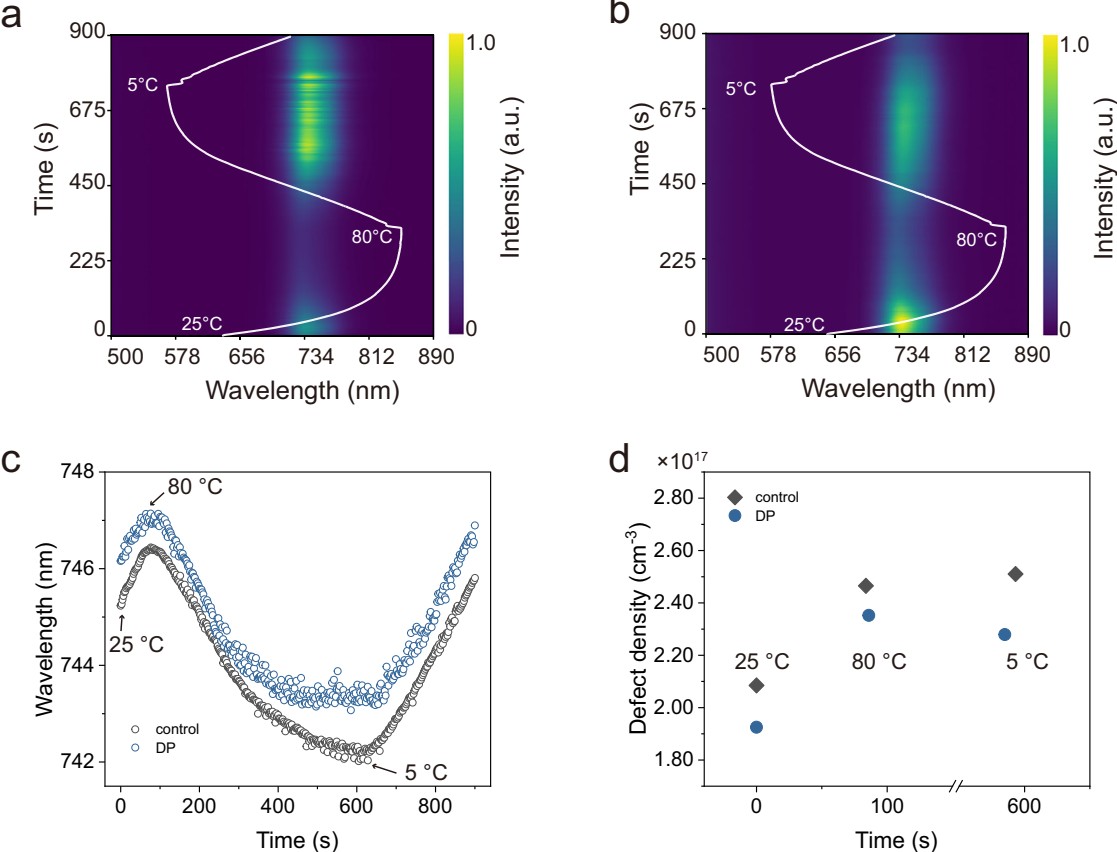

**Fig. 3 | Phase stability of WBG perovskites under rapid solar-thermal cycling.** In situ photoluminescence evolution as a function of time during thermal cycling, excited with a 450 nm laser for (**a**) control perovskite thin film and (**b**) DP perovskite thin films. The white line represents the temperature evolution, following the same time scale, with corresponding temperatures indicated. **c** Time-dependent peak position extracted from the in situ PL spectra, where the characteristic temperatures are indicated. **d** Selective defect density of control and DP perovskite thin films during thermal cycling.

elevated temperatures, increased non-radiative charge carrier recombination and structural degradation may counteract this effect, leading to non-monotonic and even decreasing $J_{SC}$ behavior.

The hysteresis index (HI), defined as $HI = \frac{PCE_{reverse} - PCE_{forward}}{PCE_{forward}}$, behaves quite differently across different types of WBG PSCs (Supplementary Fig. 15). Moreover, the HI of all types of PSCs gradually decreases over time, with the HI evolution of the DP device closely following the temperature profile (Supplementary Fig. 11). However, it remains challenging to disentangle whether ion accumulation or interfacial recombination at the perovskite/charge transport layer (CTL) interface plays the dominant role or whether both contribute jointly, to the hysteresis[33,45,46]. In addition, the hysteresis index tends to increase further at elevated temperatures, likely due to the reduced activation barrier for ion migration and the formation of trap states at the CTL interfaces[47]. To evaluate the long-time stability of these WBG PSCs under solar-thermal cycling conditions, we extend the measurement duration to 300 min (Supplementary Figs. 16–18), with selected time-resolved $J–V$ curves shown in Supplementary Fig. 19. The average PCE of control devices declines to 34% of its initial PCE, originating from the combined degradation of $V_{OC}$ (20%), $J_{SC}$ (30%), and FF (42%). In contrast, the DP devices demonstrate improved stability over prolonged solar-thermal cycling, resulting in a 46% PCE retention, attributed to a 12% degradation in $V_{OC}$, 18% in $J_{SC}$, and 37% in FF. Compared to the solar thermal cycling in the first hour, where the PCE drops significantly, further solar thermal cycling leads to only a minimal loss in both control and DP devices. Further enhancing device stability under rapid thermal cycling conditions requires holistic approaches, such as replacing commonly-used carbazole containing phosphonic acid

SAMs, which are susceptible to thermal disordering, with alternative anchoring groups (e.g., trimethoxysilane)[23], using co absorbents, bifunctional ligands[48], or superwetting overlayers to establish more rigid contact between the perovskite and the charge transport layer[49], and developing SAM bilayers or multilayer molecular contacts that harness covalent interlayer connections[9], as well as incorporating metallofullerene materials and ordered dipolar materials[34,50–52]. In addition, the respective hysteresis index is calculated and plotted against temperature and time, as shown in Supplementary Figs. 20–22, where they all follow the temperature evolution, that is, the hysteresis index reaches its maximum within one thermal cycle when operated at the highest temperature. In addition, the HI of DP devices tends to stabilize at 11.3% after an initial rapid decrease, which likely correlates with reduced ion migration and/or interfacial non-radiative recombination with dual passivation. This behavior is in stark contrast to the continuous HI evolution observed in the control devices.

## Phase and structural evolution under rapid thermal cycling

Prior to showing the structural change of WBG PSCs, we utilize in situ PL (Fig. 3a, b, Supplementary Fig. 23) to monitor the phase stability of the respective perovskite thin film under the identical conditions, particularly to simulate the initial burn-in regime (first thermal cycle). To better analyze the phase stability, the peak positions are extracted, as shown in Fig. 3c, displaying the minor changes in the peak positions before and after degradation. In other words, there is no obvious phase separation after rapid thermal cycling and light illumination and likely minimal ion migration. In addition, to quantify the impact of mobile ions on device performance at different time scales (i.e., 0 min, 15 min

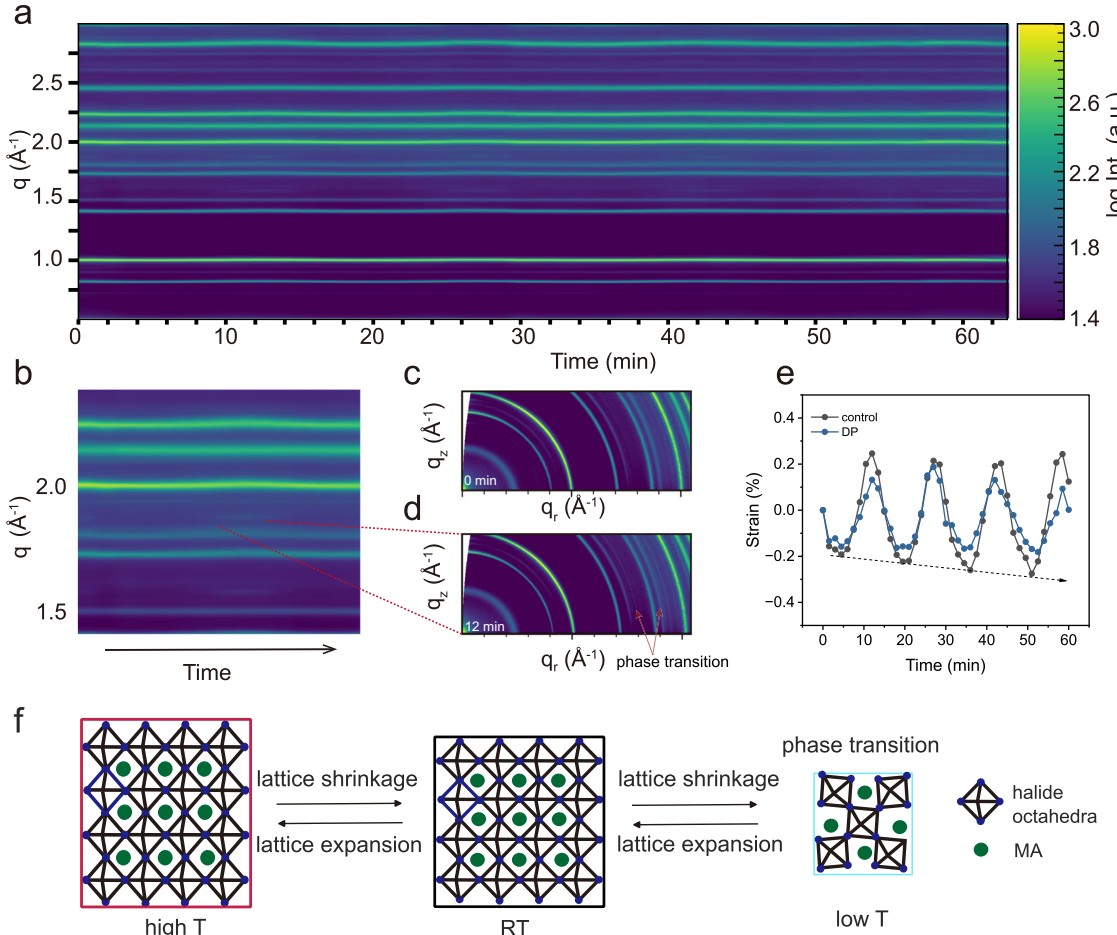

**Fig. 4 | Structure and strain evolution of WBG PSCs under rapid solarthermal cycling conditions. a** Time-dependent *operando* GIWAXS plotted retrieved from azimuthal integrated line profiles for DP PSCs under thermal cycling conditions. **b** Zoom-in time-dependent *operando* GIWAXS, showing the reversible phase transition, indicated by the red dashed line. **c** Selected reshaped 2D GIWAXS data at 0 min and (**d**) at 12 min, showing the formation of new peaks. **e** Strain evolution extracted from the evolution of $q \approx 2.0\,\text{Å}^{-1}$ of control and DP devices. The strain is calculated according to the relative $q$ shift divided by its original $q$ value at 25 °C, **f** Schematic illustration of perovskite under rapid thermal cycling, showcasing tensile and compressive strain at high temperature and low temperature, respectively, and the phase transition at low temperatures.

or 1 thermal cycle, and 60 min or 4 thermal cycles) during the aging process[46,53], we perform fast hysteresis *J–V* measurements on the DP device over a large scan speed range (0.1–800 V s⁻¹; details in "Methods"). As shown in Supplementary Fig. 24, the contribution of ionic loss remains nearly constant (-10%) before and after rapid solar-thermal cycling conditions. This also explains that in the initial burn-in regime, the rapid degradation is not attributed mainly to ion migration, but rather likely to interface loss and an increase in non-radiative recombination centers, which are related to the decrease of FF and $V_{OC}$. A similar study also demonstrated that defect density increases significantly at the perovskite/CTL interface, with this effect being more pronounced under day/night cycling conditions than under continuous illumination[54]. Furthermore, the evolution of the peak position indicates no significant bandgap shift before and after thermal cycling. Instead, a bandgap shift of 8 meV is observed within one thermal cycle for the control thin film, and 6 meV for the DP thin film. We further calculate the defect density ($N_d$) according to:

$$\Delta E = \frac{k_B T}{q} \ln\left(\frac{N_c}{N_d}\right) \qquad (3)$$

where $N_c$ refers to the effective density of states in the conductive band. Figure 3d displays the corresponding defect density of control

and DP perovskite thin films at a given temperature and time frame. It should be noted that the DP perovskite thin film demonstrates lower defect density. In addition, both perovskite thin films at 80 °C give rise to the highest defect density ($2.47 \times 10^{17}\,\text{cm}^{-3}$ for control and $2.35 \times 10^{17}\,\text{cm}^{-3}$ for DP), in accordance with the $V_{OC}$ trend in Fig. 2c. Interestingly, we notice that the DP thin film exhibits a lower defect density at 5 °C in comparison to the control thin film, corroborating its phase stability at a lower temperature. Extended in situ PL measurements further demonstrate the intrinsic stability of both control and DP thin films over two thermal cycles (Supplementary Fig. 25).

We try to understand the origin of the degradation of WBG PSCs under rapid solar-thermal cycling by correlating the device performance with their structural change (Fig. 4, Supplementary Fig. 26). To this end, we use *operando* GIWAXS, which enables concurrent tracking of structural evolution during device operation and thus provides a direct link between structural information and device behavior. It is observed that no PbI₂ peak is formed, and the perovskite peaks exhibit periodic oscillation following the temperature evolution (Fig. 4a, Supplementary Fig. 26) in all types of PSCs during device operation, that is, peak shift towards lower $q$ at elevated temperature and higher $q$ at reduced temperature, demonstrating lattice distortion over temperatures. In addition, they also undergo reversible phase transition from cubic phases at low temperature, e.g., $q \approx 1\,\text{Å}^{-1}$ and $2.0\,\text{Å}^{-1}$, to

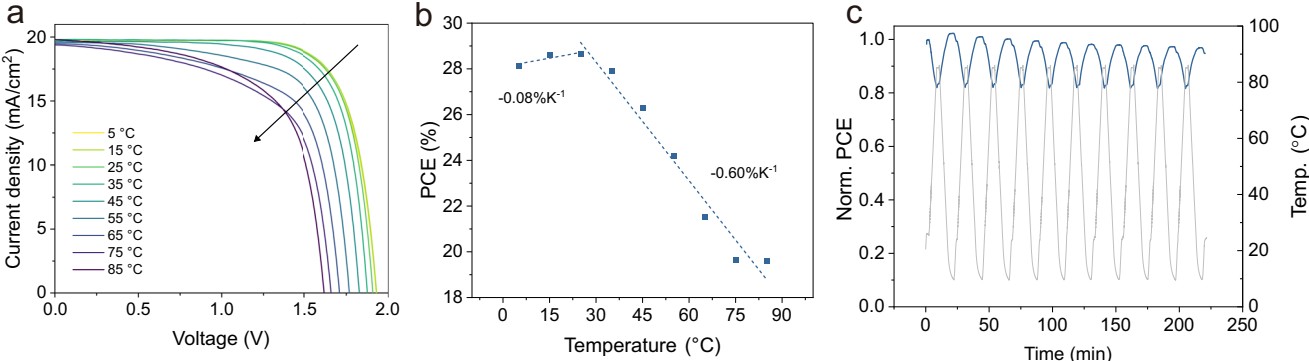

**Fig. 5 | Temperature-dependent device performance of perovskite/silicon tandem solar cells. a** Temperature-dependent $J$–$V$ measurements extracted from reverse scans. **b** PCE as a function of temperature, with the calculated temperature coefficients listed. **c** Normalized PCE evolution as a function of time under solar-thermal cycling conditions (indicated by gray curve).

tetragonal phase (β phases, $q \approx 1.65$ Å$^{-1}$ and 1.70 Å$^{-1}$, highlighted by the red dotted line, Fig. 4b–d, Supplementary Figs. 27–29)[34,55]. However, it should be pointed out that these tetragonal phases in the DP device are slightly suppressed, as indicated by the peak intensities in comparison to the control device, which also accounts for the relatively stable $J_{SC}$ against thermal cycling. To quantitatively analyze strain of PSCs operated under solar-thermal conditions, which originates from the thermal expansion coefficient mismatch between perovskite (typically $6.1 \times 10^{-5}$ K$^{-1}$) and other charge transport layers, as well as glass substrates ($3.7 \times 10^{-6}$ K$^{-1}$)[56,57], we calculate the strain as the ratio of the relative $q$ shift and its original $q$ position (Fig. 4e). The strain is oscillated expectantly with temperature evolution in the range of -0.18% (tensile strain) and 0.19% (compressive strain) in the case of the DP device, which results in the enhanced non-radiative recombination and the degradation of FF and $V_{OC}$[58], whereas the control device experiences strain in the range of −0.27% to 0.25%. The asymmetric strain of the control device might explain its large variation of FF during solar-thermal cycling conditions. It was also reported that lattice shrinkage in perovskite continuously induces deep-level defects that cannot be self-healed under day/night conditions, and this effect is expected to be more severe under accelerated thermal cycling conditions[54]. In summary, the temperature-induced strain together with the phase transition are the main driving factors that lead to the overall device failure under rapid solar-thermal cycling conditions, as illustrated in Fig. 4f. We acknowledge that further research is needed to directly measure and quantify nonradiative recombination and interfacial losses in perovskite solar cells during device operation under solar-thermal cycling conditions. Additionally, disentangling the effects of ion migration, structural changes, and charge extraction would significantly advance the understanding of the underlying physics in PSCs.

### Performance of perovskite/silicon tandem solar cells under rapid thermal cycling

Knowing the degradation behavior of single-junction WBG PSCs under rapid thermal cycling, we further extend the study on perovskite/Si tandem solar cells (detailed fabrication can be found in the Experimental Section) based on the DP device. A PCE of 28.64%, with a $V_{OC}$ of 1.908 V, $J_{SC}$ of 19.8 mA/cm$^2$, and an FF of 75.82% for the tandem solar cell at room temperature is achieved. To extract the temperature coefficients, the PCEs of TSCs are measured by statically varying the temperature between 5 and 85 °C (Fig. 5a). The device performance drops substantially at higher temperature ranges, mainly related to the degradation of FF and $V_{OC}$, which is consistent with single-junction PSCs. The calculated temperature coefficients, i.e., $\gamma_{HT}$ and $\gamma_{LT}$, are

-0.60%$_{rel}$ K$^{-1}$ and -0.08%$_{rel}$ K$^{-1}$, respectively (Fig. 5b), demonstrating enhanced stability at lower temperatures when integrating into tandem solar cells. The long-term operational stability of tandem solar cells is also examined under solar-thermal cycling conditions (Fig. 5c). Over 220 min of aging, the tandem solar cell retains 94% of its initial performance, mainly due to a reduced initial burn-in compared to single-junction PSCs. We infer that the high thermal conductivity of the silicon bottom cell facilitates efficient heat dissipation, while the tandem configuration mitigates charge carrier thermalization losses[59], collectively reducing heat accumulation and enhancing device stability under such conditions compared with single-junction PSCs. However, we also acknowledge that the differences in the layer stack, minor variations in perovskite thin film morphology arising from stoichiometry and crystallization differences on the Si bottom cells, as well as a current mismatch between the bottom and top cells, may also contribute to mitigating the initial burn-in degradation. That being said, disentangling the effects of perovskite bandgap, the phase homogeneity of perovskite, local solar spectrum, operating temperature range, and the limiting subcell in perovskite/Si tandem solar cells under constant temperature conditions remains complex[60,61], and addressing this challenge under dynamic rapid thermal cycling requires more holistic approaches combined with appropriate in situ and *operando* characterizations.

## Discussion

This work reveals the degradation behavior of WBG PSCs under rapid solar-thermal cycling conditions (temperature change rate of 10 °C/min). Similar to degradation under other external conditions, the degradation under rapid thermal cycling can be attributed to two regimes, namely initial burn-in and steady degradation, where the initial burn-in regime leads to a rapid 60% relative loss in performance. In addition, ion migration and bandgap shifts are found not to be the main culprit of driving device failure in the first regime, but rather non-radiative recombination centers in bulk and at the perovskite/CTL interface. *Operando* GIWAXS together with PL demonstrates that the temperature-induced strain, phase transition, and enhanced non-radiative recombination jointly result in the PSCs degradation, with the FF and $V_{OC}$ affecting the most. In addition, tandem solar cells retain 94% of their original PCE under solar-thermal cycling conditions over 200 min. Our work unravels the degradation behavior of WBG PSCs under thermal cycling conditions as well as that of tandem solar cells, underscoring that enhancing interfacial robustness and mitigating the effects of the initial burn-in phase are critical to improving the long-term stability of both single-junction PSCs and TSCs against thermal cycling.

## Methods

### Materials

Cesium iodide (CsI, 99.9%), formamidinium iodide (FAI, 99%), methylammonium iodide (MAI, 99%), lead iodide (PbI$_2$, 99%), lead bromide (PbBr$_2$, 99.999%), bathocuproine (BCP, 96%), chlorobenzene (anhydrous, 99.8%), dimethylformamide (DMF, anhydrous, 99.8%), dimethylsulfoxide (DMSO, anhydrous 99.9%), isopropanol (IPA, anhydrous, 99.5%), chlorobenzene (CB, anhydrous, 99.8%), and fullerene (C$_{60}$, 99.5%) were purchased from Sigma-Aldrich. [4-(3,6-Dimethyl-9H-carbazol-9-yl)butyl]phosphonic Acid (Me-4PACz, anhydrous, 98%) was purchased from Tokyo Chemical Industry. All materials were used directly without further processing if otherwise stated elsewhere.

### Single-junction WBG perovskite device fabrication

Single-junction PSCs: The ITO substrates (X07-20AC, Shangyang Tech) were ultrasonically cleaned in the sequence of detergent, DI water, ethanol, acetone, and isopropanol for 15 min each. After drying with N$_2$, the substrates were treated with O$_2$ plasma for 10 min before using. The NiO$_x$ solution (5 mg/ml) was spin-coated on ITO substrates at 3000 rpm for 30 s. The Me-4PACz solution (1 mg/mL in ethanol) was spin-coated at NiO$_x$/ITO substrates at 3000 rpm for 30 s, followed by annealing at 100 °C for 10 min. Subsequently, a 1.8 M perovskite solution (Cs$_{0.05}$MA$_{0.1}$FA$_{0.85}$PbI$_{0.77}$Br$_{0.23}$ in 1 mL mixed solvent, V$_{DMF}$: V$_{DMSO}$ = 4:1) was spin-coated on top of ITO/Me-4PACz substrates at 4000 rpm for 50 s with an acceleration of 2000 rpm/s. The anti-solvent (CB) was slowly dropped on top of the perovskite at 15 s prior to ending, followed by annealing at 100 °C for 30 min. The passivation layer was prepared by either dissolving EDAI$_2$ (1 mg) or EDAI$_2$ and F-PEAI in mixed solvent (1mg F-PEAI and 1 mg EDAI$_2$ in 1 mL IPA, stirred over night at 60 °C) and was spin-coated on top of the perovskite layer at 4000 rpm for 25 s. Then the substrates underwent 100 °C annealing for 5 min. At last, 15 nm C$_{60}$, 7 nm BCP, and 100 nm Ag were sequentially thermal evaporated on top of the perovskite layer at a pressure of 10$^{-7}$ bar. Finally, an anti-reflection foil (Mitsubishi Chemical Group) was glued from the light illumination side.

### Perovskite/Si tandem device fabrication

The silicon bottom cells were subjected to UV-ozone treatment for 5 min before NiO$_x$ modification. A NiO$_x$ film was then spin-coated as described in the single-junction fabrication process. Subsequently, the same SAM, 1.5 M perovskite, passivation layer, and C$_{60}$ deposition steps described above were applied to the Si/NiO$_x$ substrate. A 20 nm SnO$_2$ layer was deposited by atomic layer deposition (ALD) as a buffer layer. The substrate temperature was maintained at 90 °C during ALD deposition, which was carried out using Tetrakis(dimethylamino)tin(IV) (TDMASn) as the precursor at 70 °C and H$_2$O at room temperature. The pulse and purge times for TDMASn were 1 and 10.0 s with 90 sccm N$_2$, and for H$_2$O, 0.2 and 15.0 s with 90 sccm N$_2$. A total of 200 cycles were performed. Subsequently, a 45 nm IZO layer was sputtered from an IZO target through a shadow mask using 190 W power, with pure Ar and O$_2$ at 1 mTorr. An Ag finger with a thickness of 600 nm was thermally evaporated using a high-precision shadow mask. The finger width is approximately 75 µm. A 100 nm MgF$_2$ layer was thermally evaporated on top of the Ag as an anti-reflection coating.

### Device characterization

J–V measurements were recorded with a Keithley 2400 source meter under 1-sun AM 1.5 G illumination (calibrated by a Si reference cell). All single-junction devices were masked with metal aperture masks (0.05 cm$^2$) and measured under a sweep mode of reverse scan (from 1.3 V to −0.3 V) and forward scan (from −0.3 V to 1.3 V). For operando measurements under thermal cycling and for device characterization at varied temperatures, measurements were performed using both reverse (1.3 V to −0.3 V) and forward (−0.3 V to 1.3 V) voltage scans. The protocol was initiated at 25 °C and subsequently conducted at each target temperature. External quantum efficiency measurements were carried out with a Bentham PVE300-IVT system, where the LED light intensity was calibrated by the silicon and germanium diodes.[39] The QSS J–V measurements were performed by a Keithley 2400 source meter under AM 1.5 G light illumination (calibrated by a NREL-calibrated silicon reference cell). The QSS J–V measurements were 10-point measurements, where the QSS points density is set to be 99, and each point was stabilized for 45 s. For perovskite/silicon tandem solar cells, J–V measurements were carried out in the air under an LED-based solar simulator (WaveLabs Sinus 70) at room temperature. The solar simulator irradiation intensity was calibrated with a certified silicon solar cell (Fraunhofer ISE CalLab). The active area was defined by a black metal mask featuring an aperture with a precisely measured area of 1.0 cm$^2$. The devices underwent test through both reverse scans (2.1 V to −0.1 V, incrementing in 20 mV steps) and forward scans (−0.1 V to 2.1 V, with the same incremental step), conducted at a scan rate of 10 mV s$^{-1}$ and a delay time of 10 ms. For MPP tracking of TSCs, the unencapsulated devices were operated under 1-sun LED illumination (PURI materials). Fast hysteresis measurements were performed using a triangular voltage pulse from $V_{OC}$ to 0 V and back to $V_{OC}$ at different scan speeds (0.1, 1, 10, and 800 V s$^{-1}$), with the holding time at $V_{OC}$ set to five times the total scan duration[46,53].

### Materials characterization

*GIWAXS*: The thermal cycling GIWAXS data were recorded by a LAMBDA 9 M detector (X-Spectrum) with a beam energy of 11.87 keV at beamline P03 at PETRA III synchrotron (DESY, Hamburg)[62]. The data were collected with a sample-to-detector distance (SDD) of 243 mm, an exposure time of 1 s per frame, and an incidence angle of 0.6°. The SDD was calibrated with LaB$_6$ and CeO$_2$ powders with the DPDAK package and further calibrated with the ITO peak (2.132 Å$^{-1}$)[63]. Before each *operando* measurement, an irradiation damage test was performed to determine the maximum allowable exposure time per experiment, and the subsequent *operando* measurements were conducted within this time limit. The data reduction, including transformation to q-space, detector absorption, solid angle, and linecuts, was processed by the Python tool INSIGHT[64]. The setup was connected with a cooling-water system (Julabo) to exclude external heat-induced degradation, as in the previous study[65,66]. In particular, a home-built sample holder with a Peltier element is integrated in the sample holder, enabling rapid heating and cooling with each thermal cycling duration of 15 min. Considering the high temperature change rate, technical constraints, and heat dissipation, the temperature range was set to 5–85 °C. The absorption data were recorded by an ultraviolet/visible (UV/Vis) spectrometer (Perkin Elmer 35). The photoluminescence data were acquired with a fluorescence spectrometer (Perkin Elmer LS55) with an excitation wavelength of 450 nm and a slit width of 10 nm. The in situ PL was conducted with a self-constructed and mobile photoluminescence setup, which consists of a continuous laser source (Thor Labs CPS450, 450 nm) and a spectrometer (Instrument Systems CAS 140CT). The in situ measurements are conducted under thermal cycling conditions achieved by the pocket solar setup, as described above, with a laser as the light source and a time resolution of 1 s. The FTIR spectra were collected by a Broker Equinox FTIR instrument with a spectral resolution of 2 cm$^{-1}$. The atomic force microscopy measurements were carried out with an AFM instrument (Nanosurf).

### Reporting summary

Further information on research design is available in the Nature Portfolio Reporting Summary linked to this article.

## Data availability

The data generated in this study are provided in the Supplementary Information/Source Data file. Additional data are available from the corresponding author on request. Source data are provided with this paper.

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

## Acknowledgments

This work was supported by funding from the Deutsche Forschungsgemeinschaft (DFG, German Research Foundation) under Germany's Excellence Strategy—EXC 2089/1—390776260 (e-conversion), TUM.solar in the context of the Bavarian Collaborative Research Project Solar Technologies Go Hybrid (SolTech), the Center for NanoScience (CeNS), and the International Research Training Group 2022 Alberta/Technical University of Munich International Graduate School for Environmentally Responsible Functional Hybrid Materials (ATUMS). K.S., Y.L., X.J., and Z.L. acknowledge the financial support from the China Scholarship Council (CSC). We acknowledge DESY (Hamburg, Germany), a member of the Helmholtz Association HGF, for the provision of experimental facilities. Parts of this research were carried out at PETRA III. Data was collected using beamline P03 provided by DESY Photon Science. E.A. acknowledges financial support from the European Research Council (ERC) under the European Union's Horizon Europe Research and Innovation Program (INPERSPACE, Grant Agreement No. 101077006). Financial support to U.W.P by the Initiating and Networking funding of the Helmholtz Association (Solar Technology Acceleration Platform (Solar TAP)), project Zeitenwende, the program-oriented funding IV of the Helmholtz Association (Materials and Technologies for the Energy Transition, Topic 1: Photovoltaics and Wind Energy, Code: 38.01.03). R. G. acknowledges the KIT YIG-PREPRO fellowship.

## Author contributions

K.S. and R.G. conceived the idea. K.S., R.G., Q.Z., and L.F. designed the experiments and analyzed the data. K.S., Q.Z., and L.F. helped to make the devices. S.A.W. designed the thermal cycling setup. X.J., S.A.W., Z.L., Y.L., M.S., S.K.V., and S.V.R. helped to conduct *operando* GIWAXS and analysis. S.L. prepared the sketch. K.S. and R.G wrote the first manuscript draft. R.G., E.A., U.W.P., and P.M.B. supervised the project and provided resources and project administration. All authors discussed the results and commented on the manuscript.

## Funding

## Competing interests

The authors declare no competing interests.
