## [Transparent Peer Review file · Nature Communications]

Insights into the Operational Stability of Wide-Bandgap Perovskite and Tandem Solar Cells under Rapid Thermal Cycling

Corresponding Author: Dr Renjun Guo

Version 0:

Reviewer comments:

Reviewer #1

(Remarks to the Author)

In this manuscript, the authors investigated the rapid thermal cycling decay mechanisms of perovskite single-junction solar cells and tandem cells, by in-situ photoluminescence spectroscopy and in-situ GIWAXS. The conclusion drawn is that the degradation under rapid thermal cycling can be attributed to two regimes, namely initial burn-in and steady degradation. In the initial burn-in regime, the device degradation is caused by temperature-induced strain, phase transformation, and enhanced non-radiative recombination at the perovskite/CTL interface. Overall, this study is important to the community. However, more work needs to be done before the acceptance of this manuscript in Nature Communications.

Q1) In Supplementary Fig. 1a, the passivation layer changes the PL emission peak. Does this imply a change in the band gap? If so, why is this not reflected in the absorption spectrum of Supplementary Fig. 1b?

Q2) I noticed that the in-situ PL peak positions in Supplementary Fig. 3 are different from those in Supplementary Fig. 1a.

Q3) Please confirm that the parameters in Figure 1b are obviously inconsistent with the IV curve.

Q4) In Fig. 1e, more temperature conditions should be tested to verify the repeatability and accuracy of the temperature coefficients. Addition, in the acquisition of temperature coefficients and the rapid thermal cycling experiment, the authors only used three samples as statistical data, which is insufficient to support the conclusion.

Q5) The temperature coefficient of DP provided in the manuscript is $-0.42\%/K$. However, during the first cyclic heating stage (from $25^{\circ}C$ to $80^{\circ}C$), the PCE decreased by approximately 40% (corresponding to a temperature coefficient of about $-0.72\%/K$). What causes the difference in the measured temperature coefficients of devices with the same structure?

Q6) In Fig.4, in-situ GIWAXS measurements allow for a direct observation of the strain in the perovskite lattice during rapid thermal cycling. However, I have not seen additional evidence demonstrating the correlation between lattice strain/distortion and the degradation of device efficiency, and the authors should provide more supporting evidence.

Q7) In Fig.3, is the temperature measurement taken from the substrate or the perovskite film? If it is from the substrate, the actual temperature of the perovskite film requires further detection. If the temperature measurement is from the perovskite film, why is it that the temperature has been almost constant at around $80^{\circ}C$ between 100 and 250 seconds, yet the PL peak

is still undergoing a blue shift according to the data results in Figure 3c?

Q8) The discussion on the perovskite/silicon tandem solar cell section is insufficient. A more in-depth study can be conducted from aspects such as the temperature coefficient of silicon solar cells, the spectral matching of tandem cells under temperature variation, and the stress at the perovskite/silicon interface.

Reviewer #2

(Remarks to the Author)

In this work, Sun et al. report an impressive power conversion efficiency of 24.6% for single-junction wide-bandgap (1.68 eV) perovskite solar cells, representing one of the highest device performances reported to date and providing strong stimulus for further unlocking the device performance potential for perovskite/Si tandem solar cells. In addition, the authors investigate the operational stability of 1.68 eV wide bandgap perovskite solar cells under rapid solar thermal cycling conditions, a long-standing technical challenge in the field that has remained insufficiently understood, and further extend this analysis to perovskite/Si tandem devices. Given the combination of excellent device performance, valuable insights into the operational stability of both wide bandgap perovskites and perovskite/Si tandems, and the potential to broaden our understanding for extraterrestrial applications, I strongly recommend this work for publication in Nature Communications after minor revision.

Specific comments:

1. Could the authors summarize the thermal coefficients of perovskite solar cells with different bandgaps from previous publications, as this would also provide new insight of non-motocic temperature co-efficients?
2. The authors mentioned that “whereas F-PEAI is shown to form a 2D capping layer and improve the homogeneity of the perovskite/C60 layer.” Is there any evidence from, e.g., GIWAXS or PL? Also, how can one ascertain whether the molecule used for passivation can form a 2D capping layer? Can this be generalized?
3. The authors used a device configuration of ITO/NiOx/self-assembled monolayers (SAMs, Me-4PACz)/WBG PSCs (750 nm, Cs_{0.05}MA_{0.1}FA_{0.85}Pb_{0.77}Br_{0.23})/passivation layer/C60/BCP/Ag in the manuscript. How about the device performance without passivation layer under rapid solar-thermal cycling conditions? What is the effect of passivation layer?
4. Technical details regarding the operando GIWAXS measurement are missing, e.g., incidence angle. How do the authors exclude the possibility of beam-induced degradation? Since the authors calculate strain based on their q shift—which, in my opinion, is highly dependent on the sample-to-detector distance (SDD)—how can they ascertain that there was no change in SDD?
5. The authors investigated the degradation mechanism of WBG PSCs under solar thermal cycling conditions; however, including a broader discussion on potential strategies (e.g., application of metallofullerene materials and ordered dipolar materials) to enhance the stability of WBG PSCs under such conditions would significantly strengthen the manuscript.

Reviewer #3

(Remarks to the Author)

In this manuscript, the authors investigated the degradation behavior of triple-cation wide-bandgap (WBG) perovskite solar cells (PSCs) and tandem solar cells (TSCs) under rapid solar-thermal cycling. Device degradation during operation occurs in two distinct regimes: initial burn-in and steady-state degradation. This work offers valuable insights for enhancing the operational stability of WBG PSCs and TSCs, thereby facilitating their real-world applications. However, several issues remain to be addressed. Detailed comments on these aspects are provided as follows:

1. According to IEC61215:2016 protocols, the temperature range for the thermal cycling test is between -40 ± 2 °C and 85 ± 2 °C. Given this standard requirement, why do authors choose 5-85 °C instead of -40-85 °C for solar-thermal cycling tests?
2. In this manuscript, it mentioned that “We note that the thermal coefficients of single-junction PSCs and TSCs extracted from J-V measurements at constant temperatures do not reflect device performance under thermal cycling conditions.” What is the rationale for introducing the thermal coefficients, and how do they support the core conclusions of this work?
3. In comparison, the device with dual passivation using F-PEAI and EDAI₂ exhibits enhanced photovoltaic performance and thermal stability. What synergistic mechanism exists between these two molecules to contribute to such improvements?
4. In Figure 1e, how are the devices subjected to sequential testing under different temperature conditions? Additionally, within the temperature range of 5–25 °C, why does the device efficiency exhibit an increase with increasing temperature?
5. Compared with the control group, why does the optimized device with dual passivation exhibit a faster degradation rate within the 25-80 °C range?
6. After dual passivation, the device exhibits significantly reduced temperature-dependent fluctuation in FF. The authors are recommended to provide an explanation to gain deep insights into the enhanced thermal stability.
7. “As the degradation behavior is quite similar in all types of PSCs, only the evolution of photovoltaic parameters for devices with dual passivation is shown in Fig. 2b-f.” However, there is no 2f in Figure 2. The authors should supplement Figure 2f with corresponding explanations.

8. What is the temperature change rate shown in the Fig. 2b-e? It should be consistent with the change rate mentioned in the main text.

9. It mentioned that "in the initial burn-in regime, the rapid degradation is not attributed mainly to ion migration, but rather likely to interface loss and an increase in non-radiative recombination centers". Please provide related experimental characterization and in-depth analysis to verify this claim. In comparison, what are the main reasons for the performance degradation in the steady degradation regime?

10. Fig. 3d displays the corresponding defect density of control and DP perovskite thin films at a given temperature and time frame. Why is the testing time inconsistent?

11. In Supplementary Fig. 20-22, it should indicate the conditions corresponding to a and b.

12. "In single-junction PSCs, heat accumulation and hotspot formation in the absorber layers may deteriorate thermal stability and exacerbate performance loss; hence, these devices are less stable compared to tandem cells." The authors are suggested to monitor the heat accumulation within single-junction and tandem cells during thermal cycling. What is the impact of passivation layer on heat accumulation?

Version 1:

Reviewer comments:

Reviewer #1

(Remarks to the Author)

All comments have been properly addressed, and I recommend to accept this manuscript.

Reviewer #2

(Remarks to the Author)

The authors have adequately addressed all reviewer comments, and the manuscript is now suitable for acceptance.

Reviewer #3

(Remarks to the Author)

The authors have revised the work and included new data. However, the authors are suggested to provide the following suggestions for revisions to further enhance the clarity and readability of the paper. More specifically:

1. In the revised manuscript, it mentioned that " By simulating the PCE using the extracted coefficients". The authors should provide the detailed simulation process in the text. What does the red vertical line in Supplementary Fig. 14 represent? How are the two stages of the degradation process divided in figures (such as Supplementary Fig. 11 and 14)?

2. "The devices were subjected to temperature soaking at each temperature for 30 s prior to JV measurements." Can the devices and the environment reach thermal equilibrium in such a short period of time?

3. In Fig. 2, it mentioned that "showing the thermal cycling between 5 °C and 80 °C with each individual cycle of 15 min." A complete thermal cycling process (5 °C → 80 °C → 5 °C) takes 15 min. Based on this, the temperature change rate is ~10 °C/min, which is inconsistent with the change rate mentioned in the main text.

Response letter to referees

Reviewer #1 (Remarks to the Author):

In this manuscript, the authors investigated the rapid thermal cycling decay mechanisms of perovskite single-junction solar cells and tandem cells, by in-situ photoluminescence spectroscopy and in-situ GIWAXS. The conclusion drawn is that the degradation under rapid thermal cycling can be attributed to two regimes, namely initial burn-in and steady degradation. In the initial burn-in regime, the device degradation is caused by temperature-induced strain, phase transformation, and enhanced non-radiative recombination at the perovskite/CTL interface. Overall, this study is important to the community. However, more work needs to be done before the acceptance of this manuscript in Nature Communications.

Response: We thank the reviewer for the positive feedback and recommendation for publication. We are also grateful for the constructive comments, which have significantly helped improve our manuscript. In preparing our response letter, we carefully considered all suggested changes. A detailed, point-by-point response to each comment is provided below.

Q1) In Supplementary Fig. 1a, the passivation layer changes the PL emission peak. Does this imply a change in the band gap? If so, why is this not reflected in the absorption spectrum of Supplementary Fig. 1b?

Response: We thank the reviewer for pointing out this difference. We have repeated the PL measurements (Fig. R1), which show that the PL peak positions remain unchanged after passivation, indicating that the bandgap is not affected.

Fig. R1 (Supplementary Fig. 1a). Normalized photoluminescence spectra of respective perovskite thin film deposited on glass substrates.

Q2) I noticed that the in-situ PL peak positions in Supplementary Fig. 3 are different from those in Supplementary Fig. 1a.

Response: We thank the reviewer for raising this point in support of our manuscript. As noted in response to Q1, we repeated the PL measurements, confirming that passivation does not alter the perovskite bandgap, as the peak positions are consistent with those observed in the *in situ* PL results.

Fig. R2 a) Normalized PL spectra of the respective perovskite thin films. b) Normalized PL spectra extracted from the *in situ* PL measurements at room temperature.

Q3) Please confirm that the parameters in Figure 1b are obviously inconsistent with the IV curve.

Response: We thank the reviewer for pointing this out. We have corrected the typos in Figure 1b and updated the corresponding values on Page 8, Lines 1-4, in the revised manuscript. The updated version is provided here for reference.

Fig. R4 (Fig. 1b). J-V curves of the respective champion device under AM 1.5G illumination equivalent to 100 mW cm^{-2} , retrieved from the reverse scan.

Q4) In Fig. 1e, more temperature conditions should be tested to verify the repeatability and accuracy of the temperature coefficients. Addition, in the acquisition of temperature coefficients and the rapid thermal cycling experiment, the authors only used three samples as statistical data, which is insufficient to support the conclusion.

Response: We thank the reviewer for this constructive comment. As you suggested, additional temperature conditions with $5 \text{ }^\circ\text{C}$ intervals and a total 8 solar cells in the acquisition of temperature coefficients and the rapid thermal cycling experiments have been included in Figure 1e and Figure 2 in the revised manuscript, which we paste here for reference. Notably, we found that the extracted thermal coefficient depends on the temperature step size. We note that DP devices exhibit a lower temperature coefficient at higher temperatures when measured with increased precision (i.e., smaller temperature steps).

For the rapid thermal cycling experiment, we repeated the measurements once more, resulting in a total of five samples to verify repeatability and accuracy. The corresponding photovoltaic parameters have also been updated; while they remain overall similar, some slight differences are observed.

Fig. R5 (Fig. 1e). PCE as a function of temperature, where the error bar indicates the standard deviation extracted from 8 pixels.

Fig. R6 (Fig. 2): **Evolution of the device performance under solar-thermal cycling conditions.** a Schematic illustration of operando GIWAXS measurements during device operation under rapid solar-thermal cycling conditions. Normalized photovoltaic parameters as a function of time and temperature (indicated by the grey curve, showing the thermal cycling between 5 and 80 °C with each individual cycle of 15 min and a temperature change rate of

~15 °C/min). The J - V measurements were performed at 1 min intervals, and the shaded areas refer to error bars derived from the standard deviation of respective photovoltaic parameters of five samples subjected to rapid solar-thermal cycling. b Normalized PCE and temperature versus time, c normalized V_{OC} and temperature versus time, d normalized J_{SC} and temperature versus time, e normalized FF and temperature versus time.

Q5) The temperature coefficient of DP provided in the manuscript is -0.42%/K. However, during the first cyclic heating stage (from 25°C to 80°C), the PCE decreased by approximately 40% (corresponding to a temperature coefficient of about -0.72%/K). What causes the difference in the measured temperature coefficients of devices with the same structure?

Response: We are grateful for the reviewer's comment. In response to Q4, we have added additional statistics for extracting the thermal coefficients, yielding an extracted value of -0.88%/K, which is close to the one obtained from thermal cycling. The slight differences arise from the distinct measurement conditions, namely static temperature change versus continuous temperature variation. Specifically, (1) in the static case, the devices were held at the set temperature for 30 s before measuring device performance and extracting the thermal coefficients. In contrast, under thermal cycling (with a temperature change rate of 15 °C/min), the temperature varied continuously, and the devices experienced periodic strain due to the mismatch in thermal expansion coefficients between the perovskite, CTL, and ITO substrates.

(2) The time scales of electrical bias and light illumination also differed between the two conditions: under thermal cycling, electrical bias (1 min intervals) and light illumination were continuously applied, whereas under static conditions, the duration of applied bias and illumination was comparatively negligible (≤ 1 min per measurement). These differences are likely responsible for the different degradation rates of perovskite solar cells under thermal cycling compared with constant temperature conditions. We also acknowledge that electrical bias and light illumination may further contribute to device degradation.^{4, 5, 6, 7} The following statements have been added to the revised manuscript on Page 10, Lines 13-16:

The timescale of light illumination and electrical bias, together with the temperature change rate, likely collectively contribute to the different degradation rates observed in devices under rapid solar thermal cycling compared with those under static temperature conditions.

Q6) In Fig.4, in-situ GIWAXS measurements allow for a direct observation of the strain in the perovskite lattice during rapid thermal cycling. However, I have not seen additional evidence demonstrating the correlation between lattice strain/distortion and the degradation of device efficiency, and the authors should provide more supporting evidence.

Response: We thank the reviewer for the valuable comment. We carried out *operando* GIWAXS measurements designed to probe the structural evolution and device performance under actual operating conditions, rather than a perovskite film or perovskite/charge transport layer stack. Moreover, device degradation mainly originates from the loss of V_{OC} and FF, with FF loss can be attributed to the temperature-induced strain. We have further elaborated on this in the revised manuscript and have updated Figure 2a, and it reads on Page 15, Lines 12-14:

To this end, we use operando GIWAXS, which enables concurrent tracking of structural evolution during device operation and thus provides a direct link between structural information and device behavior.

Fig. R7 (Fig. 2a): Schematic illustration of *operando* GIWAXS measurements during device operation under rapid solar-thermal cycling conditions.

Q7) In Fig.3, is the temperature measurement taken from the substrate or the perovskite film? If it is from the substrate, the actual temperature of the perovskite film requires further detection. If the temperature measurement is from the perovskite film, why is it that the temperature has been

almost constant at around 80°C between 100 and 250 seconds, yet the PL peak is still undergoing a blue shift according to the data results in Figure 3c?

Response: We appreciate the reviewer's constructive comment. The temperature measurement was taken directly from the perovskite film, measured by a thermal couple. The PL signal is highly localized, and heat transfer from the perovskite surface (on the order of square centimeters) to the excitation spot (on the order of square millimeters) may introduce a time delay. In addition, like the annealing process, which typically requires some time for phase transition (e.g., in FAPbI₃), there is also a certain delay in temperature-induced peak shifts, as a sufficient amount of heat accumulation is necessary before these changes occur. This behavior explains why the PL emission peak continues to exhibit a slight blueshift over time.

Q8) The discussion on the perovskite/silicon tandem solar cell section is insufficient. A more in-depth study can be conducted from aspects such as the temperature coefficient of silicon solar cells, the spectral matching of tandem cells under temperature variation, and the stress at the perovskite/silicon interface.

Response: We thank the reviewer for raising this point. We fully agree that a more in-depth study of perovskite/silicon tandem solar cells under both constant temperature and thermal cycling conditions is necessary, and that identifying the primary causes of degradation under these conditions is of great importance to the community. However, the focus of this work is to understand the evolution of WBG PSCs under rapid solar-thermal cycling conditions. In addition, disentangling the effects of silicon bottom-cell temperature coefficients, spectrum matching of tandem devices under temperature variation, and interfacial stress at the perovskite/silicon junction is highly complex. In general, the temperature coefficients of silicon solar cells are positive because their bandgap narrows as temperature increases.⁸ When integrated into tandem devices, however, the effective temperature coefficients can be positive, negative, or a combination of both across the operating temperature range,⁹ depending on factors such as the initial current mismatch,⁹ the perovskite bandgap,¹⁰ the local spectrum,⁹ the limiting subcell,⁹ and the operating temperature,¹⁰ as the reviewer suggested. In addition, when perovskite/silicon tandem solar cells are subjected to rapid solar thermal cycling, these parameters may jointly contribute to device degradation. Given the limitations of operando techniques, along with the

overall complexity, identifying the dominant degradation mechanism remains particularly challenging. We recently discovered that the different composition and phase homogeneity of perovskites could also influence their thermal stability. Given the overall complexity of the study, it is essential to conduct a separate investigation specifically dedicated to elucidating the thermal stability of perovskite/silicon solar cells and addressing the reviewer's concerns, as this topic is also of significant importance to the field. Nevertheless, we have incorporated a discussion on this aspect into the revised manuscript, and it reads on Pages 18, Lines 19-24:

Disentangling the effects of perovskite bandgap, the phase homogeneity of perovskite, local solar spectrum, operating temperature range, and the limiting subcell in perovskite/Si tandem solar cells under constant temperature conditions remains complex,^{9, 10} and addressing this challenge under dynamic rapid thermal cycling requires more holistic approaches combined with appropriate in situ and operando characterizations.

Reviewer #2 (Remarks to the Author):

In this work, Sun et al. report an impressive power conversion efficiency of 24.6% for single-junction wide-bandgap (1.68 eV) perovskite solar cells, representing one of the highest device performances reported to date and providing strong stimulus for further unlocking the device performance potential for perovskite/Si tandem solar cells. In addition, the authors investigate the operational stability of 1.68 eV wide bandgap perovskite solar cells under rapid solar thermal cycling conditions, a long-standing technical challenge in the field that has remained insufficiently understood, and further extend this analysis to perovskite/Si tandem devices. Given the combination of excellent device performance, valuable insights into the operational stability of both wide bandgap perovskites and perovskite/Si tandems, and the potential to broaden our understanding for extraterrestrial applications, I strongly recommend this work for publication in Nature Communications after minor revision.

Response: We thank the reviewer for the positive feedback and recommendation for publication. We sincerely thank the reviewer for the constructive comments that have greatly contributed to improving our manuscript. In our response letter, we carefully considered the reviewer's suggested changes. A detailed, point-by-point response to each of the reviewers' comments is provided below.

Specific comments:

1. Could the authors summarize the thermal coefficients of perovskite solar cells with different bandgaps from previous publications, as this would also provide new insight of non-motocic temperature co-efficients?

Response: We thank the reviewer for the suggestion. We have added a table summarizing the thermal coefficients of perovskite solar cells with different bandgaps in Supplementary Table.1 and pasted it here for your reference.

Supplementary Table S1. State-of-the-art perovskite solar cells with their thermal coefficients extracted from the literature.

	Perovskite composition	Perovskite configuration	Type	Temperature range (°C)	PCE at 25°C (%)	Thermal coefficients	Ref
1	$\text{Cs}_{0.05}(\text{FA}_{0.83}\text{MA}_{0.17})\text{Pb}_{1.1}(\text{I}_{0.83}\text{Br}_{0.17})_3$	ITO/MeO-2PACz/perovskite/ C_{60} /SnO ₂ /Cu	SJ	25-85 °C	18.3%	-0.17 % _{rel} K ⁻¹	11
2	$\text{FA}_{0.79}\text{MA}_{0.16}\text{Cs}_{0.05}\text{Pb}(\text{I}_{0.83}\text{Br}_{0.17})_3$	ITO/PTAA/perovskite/LiF/ C_{60} /BCP/Ag	SJ	-20-80 °C	average ~17%	-0.36 rel %/°C	12
3	$\text{FA}_{0.75}\text{Cs}_{0.22}\text{MA}_{0.03}\text{Pb}(\text{I}_{0.82}\text{Br}_{0.15}\text{Cl}_{0.03})_3$	ITO/PTAA/perovskite/LiF/ C_{60} /BCP/Ag	SJ	-20-80 °C	average ~17%	-0.11 rel %/°C	12
4	N.A.	Perovskite/silicon tandem	TSCs	25-75 °C	25%	-0.26 % K ⁻¹	10
5	MAPbI ₃	ITO/PTAA/perovskite/ C_{60} /BCP/metal electrode	SJ	25-85 °C	16.4%	-0.13 %/°C	13
6	$\text{Cs}_{0.10}\text{FA}_{0.90}\text{Pb}(\text{I}_{0.83}\text{Br}_{0.17})_3$	FTO/TiO ₂ /perovskite/Spiro-OMeTAD/Au	SJ	0-50 °C	20%	N.A. non-monotonous	14

2. The authors mentioned that “whereas F-PEAI is shown to form a 2D capping layer and improve the homogeneity of the perovskite/ C_{60} layer.” Is there any evidence from, e.g., GIWAXS or PL? Also, how can one ascertain whether the molecule used for passivation can form a 2D capping layer? Can this be generalized?

Response: We thank the reviewer for this helpful comment. We have added the GIWAXS data for both the control and DP samples in the revised Supplementary Information and pasted it here

for reference, showing that the addition of F-PEAI leads to the formation of 2D layers ($q \approx 0.72, 0.82 \text{ \AA}^{-1}$, Supplementary Fig. 2).^{15, 16}

Fig. R8 (Supplementary Fig. 2). Pseudo-XRD data of control and DP devices, with arrows marking the 2D phase formed after F-PEAI passivation.

3. The authors used a device configuration of ITO/NiOx/self-assembled monolayers (SAMs, Me-4PACz)/WBG PSCs (750 nm, Cs_{0.05}MA_{0.1}FA_{0.85}PbI_{0.77}Br_{0.23})/passivation layer/C60/BCP/Ag in the manuscript. How about the device performance without passivation layer under rapid solar-thermal cycling conditions? What is the effect of passivation layer?

Response: The device performance without passivation (i.e., the control) behaves quite similarly under rapid solar thermal cycling conditions, also exhibiting an initial rapid degradation period followed by a steady degradation phase. However, a slight difference is observed between the control and DP devices: the control shows a 12% decrease in J_{SC} compared with 9% for the DP device. The passivation layer reduces defects and induces the formation of a low-dimensional perovskite, which can also act as a protective layer. This is supported by the calculated defect density extracted from the *in situ* PL results (Fig. 3). Nevertheless, the overall effect of passivation under rapid solar thermal cycling remains limited, as reflected in the JV results (Fig. 2 and Supplementary Fig. 12).

4. Technical details regarding the operando GIWAXS measurement are missing, e.g., incidence angle. How do the authors exclude the possibility of beam-induced degradation? Since the authors calculate strain based on their q shift—which, in my opinion, is highly dependent on the sample-to-detector distance (SDD)—how can they ascertain that there was no change in SDD?

Response: We thank the reviewer for pointing this out. We have added the technical details in the Methods section in the revised manuscript, and it reads on Page 22, Lines 24-25:

The data were collected with a sample-to-detector distance (SDD) of 243 mm, an exposure time of 1 s per frame, and an incidence angle of 0.6°.

Before each *operando* measurement, an irradiation damage test was performed to determine the maximum allowable exposure time per experiment, and the subsequent *operando* measurements were conducted within this time limit. To minimize X-ray beam–induced degradation, the sample was shifted in micrometer steps using the high-precision Hexapod at the P03 beamline, given the micrometer-sized X-ray beam. For SDD calibration, an initial calibration with CeO₂ and LaB₆ was carried out, followed by an additional calibration using the characteristic ITO peak at 2.132Å⁻¹. We therefore have included the following statements in the revised manuscript on Page 23, Lines 1-3:

Before each operando measurement, an irradiation damage test was performed to determine the maximum allowable exposure time per experiment, and the subsequent operando measurements were conducted within this time limit.

5. The authors investigated the degradation mechanism of WBG PSCs under solar thermal cycling conditions; however, including a broader discussion on potential strategies (e.g., application of metallofullerene materials and ordered dipolar materials) to enhance the stability of WBG PSCs under such conditions would significantly strengthen the manuscript.

Response: We thank the reviewer for this helpful comment. We have included a broader discussion on potential strategies for further enhancing the stability of wide-bandgap perovskite solar cells under solar-thermal cycling conditions in the revised manuscript on Page 12, Lines 3-10.

Further enhancing device stability under rapid thermal cycling conditions requires holistic approaches, such as replacing commonly-used carbazole containing phosphonic acid SAMs, which are susceptible to thermal disordering, with alternative anchoring groups (e.g., trimethoxysilane),¹⁷ using co absorbents, bifunctional ligands, or superwetting overlayers to establish more rigid contact between the perovskite and the charge transport layer,^{18, 19} and developing SAM bilayers or multilayer molecular contacts that harness covalent interlayer

connections,²⁰ as well as incorporating metallofullerene materials and ordered dipolar materials.^{21, 22, 23, 24}

Reviewer #3 (Remarks to the Author):

In this manuscript, the authors investigated the degradation behavior of triple-cation wide-bandgap (WBG) perovskite solar cells (PSCs) and tandem solar cells (TSCs) under rapid solar-thermal cycling. Device degradation during operation occurs in two distinct regimes: initial burn-in and steady-state degradation. This work offers valuable insights for enhancing the operational stability of WBG PSCs and TSCs, thereby facilitating their real-world applications. However, several issues remain to be addressed. Detailed comments on these aspects are provided as follows:

Response: We would like to thank the reviewer for the constructive comments and suggestions to further improve the quality of our manuscript. In preparing our response letter, we carefully considered all suggested changes. A detailed, point-by-point response to each comment is provided below.

1. According to IEC61215:2016 protocols, the temperature range for the thermal cycling test is between -40 ± 2 °C and 85 ± 2 °C. Given this standard requirement, why do authors choose 5-85 °C instead of -40-85 °C for solar-thermal cycling tests?

Response: We thank the reviewer for this valuable comment. We acknowledge that the IEC61215 protocol specifies that the thermal cycling test should be performed between -40 °C to 85 °C with a temperature change rate of 100 °C/h. In our work, we aimed to accelerate the solar-thermal cycling test by applying a much faster temperature change rate of 15 °C/min (900 °C/h). From a technical perspective, given this high temperature change rate, the achievable temperature range was necessarily narrowed due to the integration of our operando setup into the synchrotron facility, where both space and power were limited. These constraints impose restrictions on both the maximum attainable temperature and the temperature ramping speed. In addition, heat dissipation at such rapid temperature change rates further contributes to the temperature limitations. We have added the following statements in the revised manuscript on Page 23, Lines 9-10:

Considering the high temperature change rate, technical constraints, and heat dissipation, the temperature range was set to 5-85 °C.

2. In this manuscript, it mentioned that “We note that the thermal coefficients of single-junction PSCs and TSCs extracted from J-V measurements at constant temperatures do not reflect device performance under thermal cycling conditions.” What is the rationale for introducing the thermal coefficients, and how do they support the core conclusions of this work?

Response: We thank the reviewer for this helpful comment. By introducing thermal coefficients, we highlight their nonmonotonic behavior in SJ PSCs across different temperature regions and the contrast between SJ PSCs and TSCs. By simulating the PCE using the extracted coefficients and comparing it with the measured PCE over time and temperature (Fig. R9), we decouple degradation mechanisms from temperature-induced performance changes under rapid solar thermal cycling. Hence, the whole degradation process can be divided into two phases: the initial burn-in phase (30% PCE loss, approximately up to the first PCE recovery) and the steady degradation phase (20% PCE loss, as extracted from the two points indicated by the arrows, representing the unrecovered portion of the PCE). We have added the following statements in the manuscript, and it reads on Page 10, Lines 16-19.

By simulating the PCE using the extracted coefficients and comparing it with the measured PCE over time and temperature (Supplementary Fig.14), we decouple degradation mechanisms from temperature-induced performance changes under rapid solar thermal cycling.

Fig. R9 (Supplementary Fig. 14). Normalized PCE and simulated PCE obtained using the nonmonotonic thermal coefficients as a function of temperature and time, where the overall

degradation process can be divided into two phases: an initial burn-in phase (44% PCE loss) and a steady degradation phase (6% PCE loss, extracted from the two points indicated by the arrows).

Furthermore, as suggested, we have removed the detailed discussion of the thermal coefficients from the abstract in the revised manuscript.

3. In comparison, the device with dual passivation using F-PEAI and EDAI₂ exhibits enhanced photovoltaic performance and thermal stability. What synergistic mechanism exists between these two molecules to contribute to such improvements?

Response: We thank the reviewer for this helpful comment. Recent studies have shown that a single molecule may fail to simultaneously address both surface and interfacial recombination.²⁵ Therefore, a dual passivation strategy that mitigates both processes has been increasingly utilized. As previously reported, the combination of F-PEAI and EDAI₂ induces the formation of an n-type low-dimensional structure that facilitates interfacial charge transfer while concurrently suppressing nonradiative interfacial recombination.^{25, 26} In particular, EDAI₂ shifts the Fermi level closer to the conduction band minimum, leading to stronger n-type doping and effective field-effect passivation after treatment.²⁶ In addition, F-PEAI forms a 2D perovskite capping layer that reduces defects.²⁷ We have therefore included the following statements in the revised manuscript on Page 5, Lines 22-26, which we paste here for reference:

As demonstrated in previous work, EDAI₂ shifts the Fermi level closer to the conduction band minimum, leading to stronger n-type doping and effective field-effect passivation after treatment,^{26, 28, 29} whereas F-PEAI is shown to form a 2D capping layer (Supplementary Fig.2) and suppress the defective centers of the perovskite surface;^{27, 30} contributing to a better device performance.

4. In Figure 1e, how are the devices subjected to sequential testing under different temperature conditions? Additionally, within the temperature range of 5–25 °C, why does the device efficiency exhibit an increase with increasing temperature?

Response: We thank the reviewer for bringing this up. The devices were subjected to temperature soaking at each temperature for 30 s prior to JV measurements. Furthermore, this is precisely the point we aim to convey to the broader community. The metastability of perovskite-based devices, particularly current high-efficiency WBG PSCs and their TSCs under non-standard temperature conditions, suggests that the state-of-the-art passivation strategies may not be suitable for operation under extreme conditions or may not necessarily contribute to enhanced stability. This can be attributed to interfacial defect activation at low or elevated temperatures, ion migration, phase transformation, and other related dynamic processes. Therefore, developing dedicated passivation strategies to address these challenges is essential. The following statement has been added to the revised manuscript, and it reads on Page 9, Lines 23-26:

Overall, the metastability of high-efficiency WBG PSCs under non-standard temperature conditions suggests that current passivation strategies may not be suitable for operation under extreme conditions or may not necessarily contribute to enhanced stability, highlighting the need to develop dedicated strategies specifically aimed at improving stability.

5. Compared with the control group, why does the optimized device with dual passivation exhibit a faster degradation rate within the 25-80°C range?

Response: We thank the reviewer for raising this point. Indeed, dual passivation can largely enhance device efficiency, and the stability of thin film and device under rapid solar-thermal cycling conditions, particularly with low temperature range, which attributes to the less defect densities and strain (see Question 6). However, previous studies have shown that the dimensionality of 2D perovskites formed at the 3D/2D heterojunction can collapse and reform depending on the aging conditions and their bulky organic spacers,^{31, 32, 33} which can trigger redox reactions that reduce charge transport mobility. This collapse is typically more pronounced under photo-thermal aging at higher temperatures (e.g., 85 °C). Therefore, we infer that 2D/3D passivation is more effective at lower temperatures than at higher temperatures. We have included the following statement in the revised manuscript on Page 9, Lines 7-11:

Notably, the temperature coefficients in the 5-25 °C range (γ_{LT}) and the 25-80°C (γ_{HT}) are largely different, where PSCs with dual passivation demonstrate better stability at lower temperatures. At higher temperatures (above room temperature), however, DP devices suffer

from a faster degradation, likely due to dimensionality collapse, penetration of bulky organic cation into the 3D perovskite layer within 2D/3D heterostructure, which hinders charge transfer.^{31, 32, 33}

6. After dual passivation, the device exhibits significantly reduced temperature-dependent fluctuation in FF. The authors are recommended to provide an explanation to gain deep insights into the enhanced thermal stability.

Response: We thank the reviewer for this helpful comment. As suggested, we have included the strain analysis in Figure 4e and extracted the evolution of the series resistance, demonstrating that the control sample experiences larger strain and higher series resistance under solar thermal cycling conditions.

The fill factor can be expressed as:

$$FF = \frac{v_{OC} - \ln(v_{OC} + 0.72)}{v_{OC} + 1} \left(1 - R_s \frac{J_{SC}}{V_{OC}}\right)$$

in the limit of infinitely large shunting resistance, where R_s refers to the series resistance.^{34, 35, 36}

First, *in situ* PL results (Fig. 3) show that DP devices exhibit lower defect densities compared with the control devices under rapid solar thermal cycling. Second, GIWAXS strain analysis (Fig. 4e) reveals that the DP devices exhibit smaller strain fluctuations (approximately from -0.18% to 0.19%) compared to the control devices (approximately from -0.25% to 0.27%). The control devices also exhibit a gradual increase in tensile strain (indicated by the arrow in the figure), which may contribute to the large FF fluctuations observed. To further corroborate these findings, we have plotted the series resistance as a function of time (Supplementary Fig. 13) and included it in the revised SI, showing that the control devices display significantly larger fluctuations compared with the DP devices. In addition, shunt resistance may also contribute to FF losses, and variations in device behavior (as reflected by their standard deviations) likely explain the larger FF variations observed in the control devices.

Fig. R10 (Fig. 4e). Strain evolution extracted from the evolution of control and DP devices at $q \approx 2.0 \text{ \AA}^{-1}$. The strain is calculated according to the relative q shift divided by its original q value at 25°C .

Fig. R11 (Supplementary. Fig.13). Normalized series resistance of control and DP device as a function of time.

Hence, the following statement has been added in the revised manuscript on Page 10, Lines 10-13:

The FF fluctuations in control devices (characterized by the series resistance in the limit of infinitely large shunting resistance, Supplementary Fig.13) can be correlated with the high defect densities and the asymmetric strain experienced during rapid solar-thermal cycling, which will be discussed in the following sections.

7. “As the degradation behavior is quite similar in all types of PSCs, only the evolution of photovoltaic parameters for devices with dual passivation is shown in Fig. 2b-f.” However, there is no 2f in Figure 2. The authors should supplement Figure 2f with corresponding explanations.

Response: We appreciate the reviewer for pointing this out. We have corrected this in the revised manuscript, where the evolution of the hysteresis index is now shown in Supplementary Figure 11, and it now reads on Page 10, Lines 3-5:

As the degradation behavior is quite similar in all types of PSCs, only the evolution of photovoltaic parameters for devices with dual passivation is shown in Fig. 2b-e and Supplementary Fig.11.

Fig. R12 (Supplementary Fig. 11). Normalized hysteresis index of DP device and temperature versus time.

8. What is the temperature change rate shown in the Fig. 2b-e? It should be consistent with the change rate mentioned in the main text.

Response: We thank the reviewer for raising this question. We have now added the temperature change rate (15 °C/min) in the Figure caption, which is consistent with the change rate mentioned

in the main text.

9. It mentioned that “in the initial burn-in regime, the rapid degradation is not attributed mainly to ion migration, but rather likely to interface loss and an increase in non-radiative recombination centers” . Please provide related experimental characterization and in-depth analysis to verify this claim. In comparison, what are the main reasons for the performance degradation in the steady degradation regime?

Response: We thank the reviewer for raising this question. As suggested, we conducted fast hysteresis measurements, in which the device was scanned over a wide range of scan speed (10^{-3} - 10^3 Vs^{-1}), enabling the extraction of the ionic loss originating from the redistribution of mobile ions.^{37, 38}

The experimental details are as follows:

Fast hysteresis measurements were conducted before and after solar-thermal cycling at different time intervals (i.e., 0 min, 15 min or 1 thermal cycle, and 60 min or 4 thermal cycles) to investigate the effect of ion migration on device performance. As shown in the figure, the ionic loss before and after rapid-solar thermal cycling (1h) remains relatively constant (~10%), demonstrating that ionic loss is not the main driving factor of degradation. Instead, interfacial loss and non-radiative recombination centers (as indicated by the defect densities in the *in situ* PL data) are more likely responsible for the degradation, as further evidenced by the significant loss of FF and V_{OC} . We have now added the corresponding figure in the revised SI and pasted it here for reference. Accordingly, we have revised the statements in the manuscript, and it reads on Page 14, Lines 1-8:

In addition, to quantify the impact of mobile ions on device performance at different time scales (i.e., 0 min, 15 min, or 1 thermal cycle, and 60 min or 4 thermal cycles) during the aging process, we perform fast hysteresis JV measurements on DP device over a large scan speed range (0.1-800 V s^{-1} ; details in Methods). As shown in Supplementary Fig. 24, the contribution of ionic loss remains nearly constant (~10%) before and after rapid solar-thermal cycling conditions. This also explains that in the initial burn-in regime, the rapid degradation is not attributed mainly to ion migration, but rather likely to interface loss and an increase in non-radiative recombination centers, which are related to the decrease of FF and V_{OC} .

Fig. R13 (Supplementary Fig. 24). a) The absolute PCE at fast and slow scan speeds versus aging time under rapid solar-thermal cycling conditions. b) V_{oc} at fast and slow scan speeds versus aging time. c) J_{sc} at fast and slow scan speeds versus aging time. d) FF at fast and slow scan speeds versus aging time, and e) Exemplary JV curves measured at fast and slow scan speeds at different aging times.

10. Fig. 3d displays the corresponding defect density of control and DP perovskite thin films at a given temperature and time frame. Why is the testing time inconsistent?

Response: We thank the reviewer for this helpful comment. Since the majority of degradation occurs during the initial rapid degradation stage, within the first 15 minutes or during the first thermal cycle, we aimed to simulate this period using an identical timescale. In addition, Figure 3d presents representative data points at the specified times and temperatures. We have added the following statement in the revised manuscript on Page 13, Lines 13-16:

Prior to showing the structural change of WBG PSCs, we utilize in situ PL (Fig. 3a,b, and Supplementary Fig.23) to monitor the phase stability of the respective perovskite thin film, particularly to simulate the initial burn-in regime under the conditions of rapid thermal cycling and continuous light illumination.

As suggested, we also extended the measurement duration to up to two thermal cycles and have added it here and in Supplementary Fig.25:

Fig. R14 (Supplementary Fig. 25): *In situ* photoluminescence evolution as a function of time over two thermal cycles, excited with a 450 nm laser for a control perovskite thin film and b DP perovskite thin film.

11. In Supplementary Fig. 20-22, it should indicate the conditions corresponding to a and b.

Response: We thank the reviewer for raising this point. We have added the conditions corresponding to panels a and b in the captions of Supplementary Figs. 27–29. The updated version is also pasted here for reference.

Fig. R15 (Supplementary Fig. 27). Selected reshaped 2D GIWAXS data of the control perovskite solar cells at a) 25 °C and b) at 5 °C within one thermal cycle, where the red arrows indicate the phase transition.

Fig. R16 (Supplementary Fig. 28). Selected reshaped 2D GIWAXS data of the EDAl₂ perovskite solar cells at a) 25 °C and b) at 5 °C within one thermal cycle, where the red arrows indicate the phase transition.

Fig. R17 (Supplementary Fig. 29). Selected reshaped 2D GIWAXS data of data DP perovskite solar cells at a) 25 °C and b) at 5 °C within one thermal cycle, where the red arrows indicate the phase transition.

12. “In single-junction PSCs, heat accumulation and hotspot formation in the absorber layers may deteriorate thermal stability and exacerbate performance loss; hence, these devices are less stable compared to tandem cells.” The authors are suggested to monitor the heat accumulation within single-junction and tandem cells during thermal cycling. What is the impact of passivation layer on heat accumulation?

Response: We thank the reviewer for raising this point. We agree that the claim “heat accumulation and hotspot formation in the absorber ...” may be misleading and have therefore removed it. In our case, the rapid heating and cooling rates indicate active thermal dissipation and efficient heat exchange, suggesting that the effect of heat accumulation should be minimal. For this reason, we have removed the statement from the manuscript. In addition, since the passivation layer is only a few tens of nanometers thick, much thinner than the perovskite active layer (several hundred nanometers) and the TCO layer (millimeter scale), its contribution to heat accumulation is expected to be minimal. However, as discussed in Q5, the dual passivation results in faster degradation, likely due to dimensionality collapse and the penetration of bulky organic cations into the 3D perovskite layer within the 2D/3D heterostructure, which hinders charge transfer.^{31, 32, 33} Also, our results demonstrate that suitable passivation strategies should be explored to enhance the stability of PSCs under extreme conditions, instead of targeting PCE improvement alone. Moreover, the tandem configuration enables more efficient utilization of the solar spectrum than single-junction PSCs, as high-energy photons are absorbed by the wide-bandgap top cell, thereby generating a higher voltage and reducing thermalization losses. Consequently, thermal accumulation within the absorber is alleviated relative to single-junction devices.

We have included the discussion in the revised manuscript, and it reads on Page 18, Lines 12-15:

We infer that the high thermal conductivity of the silicon bottom cell facilitates efficient heat dissipation, while the tandem configuration mitigates carrier thermalization losses,³⁹ collectively reducing heat accumulation and enhancing device stability under such conditions compared with single-junction PSCs.

References

1. Tennyson EM, Doherty TA, Stranks SD. Heterogeneity at multiple length scales in halide perovskite semiconductors. *Nature Reviews Materials* **4**, 573-587 (2019).
2. Yoon SJ, Kuno M, Kamat PV. Shift happens. How halide ion defects influence photoinduced segregation in mixed halide perovskites. *ACS Energy Letters* **2**, 1507-1514 (2017).
3. Shao Z, *et al.* Temperature-dependent photoluminescence of Co-evaporated MAPbI₃ ultrathin films. *Results in Physics* **34**, 105326 (2022).
4. Khenkin MV, Anoop K, Katz EA, Visoly-Fisher I. Bias-dependent degradation of various solar cells: lessons for stability of perovskite photovoltaics. *Energy & Environmental Science* **12**, 550-558 (2019).
5. Domanski K, Alharbi EA, Hagfeldt A, Grätzel M, Tress W. Systematic investigation of the impact of operation conditions on the degradation behaviour of perovskite solar cells. *Nature Energy* **3**, 61-67 (2018).
6. Motti SG, Patel JB, Oliver RD, Snaith HJ, Johnston MB, Herz LM. Phase segregation in mixed-halide perovskites affects charge-carrier dynamics while preserving mobility. *Nature Communications* **12**, 6955 (2021).
7. Fang Z, Nie T, Liu S, Ding J. Overcoming phase segregation in wide-bandgap perovskites: from progress to perspective. *Advanced Functional Materials* **34**, 2404402 (2024).
8. Green MA. Self-consistent optical parameters of intrinsic silicon at 300 K including temperature coefficients. *Solar Energy Materials and Solar Cells* **92**, 1305-1310 (2008).
9. Babics M, Bristow H, Pininti AR, Allen TG, De Wolf S. Temperature coefficients of perovskite/silicon tandem solar cells. *ACS Energy Letters* **8**, 3013-3015 (2023).
10. Aydin E, *et al.* Interplay between temperature and bandgap energies on the outdoor performance of perovskite/silicon tandem solar cells. *Nature Energy* **5**, 851-859 (2020).
11. Jošt M, *et al.* Perovskite solar cells go outdoors: field testing and temperature effects on energy yield. *Advanced energy materials* **10**, 2000454 (2020).
12. Moot T, *et al.* Temperature coefficients of perovskite photovoltaics for energy yield calculations. *ACS Energy Letters* **6**, 2038-2047 (2021).
13. Deng Y, Van Brackle CH, Dai X, Zhao J, Chen B, Huang J. Tailoring solvent coordination for high-speed, room-temperature blading of perovskite photovoltaic films. *Science advances* **5**, eaax7537 (2019).

14. Tress W, *et al.* Performance of perovskite solar cells under simulated temperature-illumination real-world operating conditions. *Nature energy* **4**, 568-574 (2019).
15. Bouduban ME, *et al.* Crystal orientation drives the interface physics at two/three-dimensional hybrid perovskites. *The journal of physical chemistry letters* **10**, 5713-5720 (2019).
16. Zhou Q, *et al.* Dually-passivated perovskite solar cells with reduced voltage loss and increased super oxide resistance. *Angewandte Chemie International Edition* **60**, 8303-8312 (2021).
17. Tang H, *et al.* Reinforcing self-assembly of hole transport molecules for stable inverted perovskite solar cells. *Science* **383**, 1236-1240 (2024).
18. He R, *et al.* Improving interface quality for 1-cm² all-perovskite tandem solar cells. *Nature* **618**, 80-86 (2023).
19. Zheng X, *et al.* Co-deposition of hole-selective contact and absorber for improving the processability of perovskite solar cells. *Nature Energy* **8**, 462-472 (2023).
20. Dong B, *et al.* Self-assembled bilayer for perovskite solar cells with improved tolerance against thermal stresses. *Nature Energy* **10**, 342-353 (2025).
21. Li G, *et al.* Highly efficient pin perovskite solar cells that endure temperature variations. *Science* **379**, 399-403 (2023).
22. Wang W, *et al.* Thermal Cross-Linking Hole-Transport Self-Assembled Monolayers for Perovskite Solar Cells. *ACS Energy Letters* **10**, 2250-2258 (2025).
23. Li Z, *et al.* In-Situ Cross-Linked Polymers for Enhanced Thermal Cycling Stability in Flexible Perovskite Solar Cells. *Angewandte Chemie International Edition* **64**, e202421063 (2025).
24. Lin Y, *et al.* A Nd@C82-polymer interface for efficient and stable perovskite solar cells. *Nature*, 1-3 (2025).
25. Liu C, *et al.* Bimolecularly passivated interface enables efficient and stable inverted perovskite solar cells. *Science* **382**, 810-815 (2023).
26. Jiang X, *et al.* Surface heterojunction based on n-type low-dimensional perovskite film for highly efficient perovskite tandem solar cells. *National Science Review* **11**, nwae055 (2024).
27. Wang Y, *et al.* Homogenized contact in all-perovskite tandems using tailored 2D perovskite. *Nature* **635**, 867-873 (2024).

28. Liu J, *et al.* Perovskite/silicon tandem solar cells with bilayer interface passivation. *Nature* **635**, 596-603 (2024).
29. Chen H, *et al.* Regulating surface potential maximizes voltage in all-perovskite tandems. *Nature* **613**, 676-681 (2023).
30. Chen H, *et al.* Quantum-size-tuned heterostructures enable efficient and stable inverted perovskite solar cells. *Nature Photonics* **16**, 352-358 (2022).
31. Peng Z, *et al.* Revealing degradation mechanisms in 3D/2D perovskite solar cells under photothermal accelerated ageing. *Energy & Environmental Science* **17**, 8313-8324 (2024).
32. Xu X, Zhang Z, Liu T, Zhu P, Zhang Z, Xing G. Suppressing the penetration of 2D perovskites for enhanced stability of perovskite solar cells. *Journal of Materials Chemistry A* **13**, 12097-12103 (2025).
33. Fiorentino F, Albaqami MD, Poli I, Petrozza A. Thermal-and light-induced evolution of the 2D/3D interface in lead-halide perovskite films. *ACS Applied Materials & Interfaces* **14**, 34180-34188 (2021).
34. Razzaq A, Ullah A, Subbiah AS, De Wolf S. Practical fill factor limits for perovskite solar cells. *ACS Energy Letters* **9**, 5635-5638 (2024).
35. Ma C, Park N-G. A realistic methodology for 30% efficient perovskite solar cells. *Chem* **6**, 1254-1264 (2020).
36. Wang Y, Akel S, Klingebiel B, Kirchartz T. Hole Transporting Bilayers for Efficient Micrometer-Thick Perovskite Solar Cells. *Advanced Energy Materials* **14**, 2302614 (2024).
37. Thiesbrummel J, *et al.* Ion-induced field screening as a dominant factor in perovskite solar cell operational stability. *Nature Energy* **9**, 664-676 (2024).
38. Le Corre VM, *et al.* Quantification of efficiency losses due to mobile ions in perovskite solar cells via fast hysteresis measurements. *Solar RRL* **6**, 2100772 (2022).
39. Tong J, Jiang Q, Zhang F, Kang SB, Kim DH, Zhu K. Wide-bandgap metal halide perovskites for tandem solar cells. *ACS Energy Letters* **6**, 232-248 (2020).

Response letter to referees

Reviewer #1 (Remarks to the Author):

All comments have been properly addressed, and I recommend to accept this manuscript.

Response: We thank the reviewer for the constructive suggestions for the first round and recommendation for publication. We are pleased that our revisions and clarifications have addressed your concerns.

Reviewer #2 (Remarks to the Author):

The authors have adequately addressed all reviewer comments, and the manuscript is now suitable for acceptance.

Response: We thank the reviewer for your endorsement for our work and recommendation for publication.

Reviewer #3 (Remarks to the Author):

The authors have revised the work and included new data. However, the authors are suggested to provide the following suggestions for revisions to further enhance the clarity and readability of the paper. More specifically:

We are also grateful for the constructive comments, which have significantly helped improve our manuscript. In preparing our response letter, we carefully considered all suggested changes. A detailed, point-by-point response to each comment is provided below.

1. In the revised manuscript, it mentioned that " By simulating the PCE using the extracted coefficients". The authors should provide the detailed simulation process in the text. What does

the red vertical line in Supplementary Fig. 14 represent? How are the two stages of the degradation process divided in figures (such as Supplementary Fig. 11 and 14)?

Response: We would like to thank the reviewer for the constructive comments and suggestions to further improve the quality of our manuscript.

The simulation was based on the initial room-temperature PCE and the measured temperature coefficients for 5–25 °C and 25–85 °C. These coefficients were applied to the experimental temperature profile, assuming no additional performance losses. We have added the following statements in the figure caption of Supplementary Fig.14 and it reads on Page 15:

The simulation is carried out to model the expected behavior under the experimental temperature profile, assuming no concurrent performance loss (i.e., degradation unrelated to reversible thermal effects). The model is built using the initial PCE recorded at room temperature and the extracted temperature coefficients for the specified ranges (5–25 °C and 25–85 °C), which are then applied across the measured temperature profile.

In addition, the red vertical line is intended to separate the two degradation regimes. Different stages are defined by the point in time where the solar cell's measured behavior first aligned with both the temperature profile and the simulated J-V curve. This transition point is then consistently applied to all photovoltaic parameters. We have therefore revised the dash lines in Supplementary Fig.11 as well as in Fig. 2b-d to improve the readability of our manuscript.

Fig. 2: Evolution of the device performance under solar-thermal cycling conditions. a Schematic illustration of *operando* GIWAXS measurements during device operation under rapid solar-thermal cycling conditions. Normalized photovoltaic parameters as a function of time and temperature (indicated by the grey curve, showing the thermal cycling between 5 and 80 °C with each individual cycle of 15 min and a temperature change rate of ~ 10 °C/min). The J - V measurements were performed at 1 min intervals, and the shaded areas refer to error bars derived from the standard deviation of respective photovoltaic parameters of five pixels subjected to rapid solar-thermal cycling. b Normalized PCE and temperature versus time, c normalized V_{OC} and temperature versus time, d normalized J_{SC} and temperature versus time, e. normalized FF and temperature versus time.

2. "The devices were subjected to temperature soaking at each temperature for 30 s prior to JV measurements." Can the devices and the environment reach thermal equilibrium in such a short period of time?

Response: We thank the reviewer for this insightful question. In a typical perovskite device, the time to reach a thermal equilibrium is governed by the thermal time constant (τ). Given that the substrate (TCO/glass) thickness (0.7 mm) far exceeds the active layer thickness (less than 1 μm), we neglect the thin films and focus on the substrate.

The time constant can be calculated as:

$$\tau = \frac{L^2}{\alpha}$$

where L is the thickness (0.7 mm), and α is the thermal diffusivity, defined as $\alpha = k/(\rho C_p)$. Using the values of $k = 1.0 \text{ W}/(\text{m}\cdot\text{K})$, $\rho = 2500 \text{ kg}/\text{m}^3$, $C_p = 840 \text{ J}/(\text{kg}\cdot\text{K})$, the calculated time constant is approximately 1 s, leading to a time to reach of equilibrium (assuming a temperature change from 25 $^{\circ}\text{C}$ to 85 $^{\circ}\text{C}$) of roughly 3 s. Hence, the established 30 s soaking period is more than sufficient to ensure the device has reached thermal equilibrium prior to J-V measurements.

3. In Fig. 2, it mentioned that "showing the thermal cycling between 5 $^{\circ}\text{C}$ and 80 $^{\circ}\text{C}$ with each individual cycle of 15 min." A complete thermal cycling process (5 $^{\circ}\text{C} \rightarrow 80 \text{ }^{\circ}\text{C} \rightarrow 5 \text{ }^{\circ}\text{C}$) takes 15 min. Based on this, the temperature change rate is $\sim 10 \text{ }^{\circ}\text{C}/\text{min}$, which is inconsistent with the change rate mentioned in the main text.

Response: We thank the reviewer for pointing out this oversight. The temperature change rate has been corrected to $\sim 10 \text{ }^{\circ}\text{C}/\text{min}$ in both the main text and the corresponding figure caption.

The authors have revised the work and included new data. However, the authors are suggested to provide the following suggestions for revisions to further enhance the clarity and readability of the paper. More specifically:

1. In the revised manuscript, it mentioned that " By simulating the PCE using the extracted coefficients". The authors should provide the detailed simulation process in the text. What does the red vertical line in Supplementary Fig. 14 represent? How are the two stages of the degradation process divided in figures (such as Supplementary Fig. 11 and 14)?
2. "The devices were subjected to temperature soaking at each temperature for 30 s prior to JV measurements." Can the devices and the environment reach thermal equilibrium in such a short period of time?
3. In Fig. 2, it mentioned that "showing the thermal cycling between 5 °C and 80 °C with each individual cycle of 15 min." A complete thermal cycling process (5 °C → 80 °C → 5 °C) takes 15 min. Based on this, the temperature change rate is ~10 °C/min, which is inconsistent with the change rate mentioned in the main text.

Fig. 2: Evolution of the device performance under solar-thermal cycling conditions. a Schematic illustration of *operando* GIWAXS measurements during device operation under rapid solar-thermal cycling conditions. Normalized photovoltaic parameters as a function of time and temperature (indicated by the grey curve, showing the thermal cycling between 5 and 80 °C with each individual cycle of 15 min and a temperature change rate of ~15 °C/min). The *J-V* measurements were performed at 1 min intervals, and the shaded areas refer to error bars derived from the standard deviation of respective photovoltaic parameters of five pixels subjected to rapid solar-thermal cycling. **b** Normalized *PCE* and temperature *versus* time, **c** normalized *V_{OC}* and temperature *versus* time, **d** normalized *J_{SC}* and temperature *versus* time, **e**. normalized *FF* and temperature *versus* time.